# Specificity of the Hox member Deformed is determined by transcription factor levels and binding site affinities

Pedro B. Pinto[1,2], Katrin Domsch [1], Xuefan Gao[1], Michaela Wölk[1,3], Julie Carnesecchi [1,2] & Ingrid Lohmann [1] ✉

Hox proteins have similar binding specificities in vitro, yet they control different morphologies in vivo. This paradox has been partially solved with the identification of Hox low-affinity binding sites. However, anterior Hox proteins are more promiscuous than posterior Hox proteins, raising the question how anterior Hox proteins achieve specificity. We use the *AP2x* enhancer, which is activated in the maxillary head segment by the Hox TF Deformed (Dfd). This enhancer lacks canonical Dfd-Exd sites but contains several predicted low-affinity sites. Unexpectedly, these sites are strongly bound by Dfd-Exd complexes and their conversion into optimal Dfd-Exd sites results only in a modest increase in binding strength. These small variations in affinity change the sensitivity of the enhancer to different Dfd levels, resulting in perturbed *AP-2* expression and maxillary morphogenesis. Thus, Hox-regulated morphogenesis seems to result from the co-evolution of Hox binding affinity and Hox dosage for precise target gene regulation.

Hox proteins are highly conserved homeodomain transcription factors (TFs) with key roles in development[1-4]. Hox TFs specify segment identity along the anterior-posterior axis and are responsible for the morphological diversity of structures along this body plan[1]. Due to their biological importance and their conservation throughout metazoans, a huge body of work has been directed to search for targets of different Hox TFs[5] and, more importantly, to determine how the different Hox TFs activate their target genes[6]. This latter question is particularly important as Hox TFs share a highly conserved DNA-binding domain, the homeodomain[7,8]. Contrary to their highly specific functions in vivo, Hox TFs recognize similar binding sequences in vitro[8]. This so-called Hox paradox has been partially resolved with the discovery of two conserved three-amino acid loop extension (TALE) homeodomain TFs: Extradenticle (Exd) and Homothorax (Hth) in invertebrates and Pbx and Meis in vertebrates, respectively[6]. These cofactors are required for Hox proteins to bind and activate specific Hox-regulated *cis*-regulatory modules (CRMs).

They modulate the Hox recognition of specific sequences with different Hox–Exd complexes recognizing different binding sequences with different affinities.

Systematic evolution of ligands by exponential enrichment (SELEX) studies of all *Drosophila* Hox TFs characterized groups of binding sites that are preferentially recognized by the different Hox–Exd complexes[9]. Accordingly, Hox complexes are organized in three classes according to binding site preferences. Class 1 binding sites, with a nTGATTGATnnn core sequence, are bound preferentially by Labial (Lb) and Proboscipedia (Pb), while class 2 sequences with a nTGATTAATnnn core sequence are preferred targets of Deformed (Dfd) and Sex comb reduced (Scr). Class 3 Hox–Exd complexes are composed of Antennapedia (Antp), Ultrabithorax (Ubx), Abdominal-A (Abd-A) and Abdominal B (Abd-B) and show a higher affinity towards nTGATTTATnnn core sequences. These results provide a generic framework to understand how Hox TFs recognize their specific target genes. However, they also showed

[1]Department of Developmental Biology and Cell Networks - Cluster of Excellence, Heidelberg University, Centre for Organismal Studies (COS) Heidelberg, Heidelberg, Germany. [2]Present address: Institut de Genomique Fonctionnelle de Lyon, Univ Lyon, CNRS UMR 5242, Ecole Normale Superieure de Lyon, Universite Claude Bernard Lyon 1, Lyon, France. [3]Present address: Friedrich Miescher Institute for Biomedical Research (FMI), Basel, Switzerland. ✉e-mail: ingrid.lohmann@cos.uni-heidelberg.de

that Hox proteins are able to recognize a diverse number of binding sites with different affinities.

The importance of affinity with which TFs bind to CRMs has been the subject of several studies and shown to be important for the fine-tuning of gene expression[10–13]. The study of the *shavenbaby* (*svb*) enhancers *E3N* and *7H*, which are regulated by the Hox TF Ubx, highlighted the importance of binding site diversity and affinity for Hox specificity[12]. These regulatory regions contain non-canonical low-affinity class 3 sites necessary for the specific expression of *svb* in the Ubx domain. Since Hox–Exd class 2 complexes (Dfd and Scr) are the most promiscuous of the Hox–Exd complexes, able to recognize class 1 and 3 sites[9], the segment-specific expression of *svb* relies on a trade-off between activity and specificity. Optimization of the Ubx–Exd binding sites in the *E3N* and *7H* enhancers resulted in the loss of specificity due to the activation of these enhancers by Scr–Exd complexes in the anterior region of the *Drosophila* embryo.

Overall, these results suggested that high-affinity binding sites are present when Hox specificity is secondary, as target genes are activated by all Hox proteins. The study of the *Drosophila ventral veinless* (*vvl*) enhancer *vvl1 + 2* supported this observation[14]. This CRM contains high-affinity class 2 and class 3 Hox–Exd sites and is activated across the anterior–posterior axis by different Hox TFs. The existence of Hox–Exd low-affinity binding sites provides an explanation of how posterior Hox–Exd complexes activate their target genes in a tissue-specific manner while preventing activation by anterior Hox–Exd complexes, capable of binding a wider range of binding sites. However, it also suggests that affinity might play a different role in the activation of anterior Hox target genes.

In this work, we studied the embryonic maxillary enhancer *AP2x* of the *Drosophila* gene *AP-2/tfAP-2*[15]. This enhancer directs the expression of the TF encoding gene *AP-2* in a specific domain of the maxillary segment, under the control of Dfd. *AP2x* lacks class 2 canonical Dfd–Exd sites but contains instead several predicted low-affinity/non-canonical Dfd–Exd sites. Thus, the *AP2x* enhancer is an excellent model to address Hox-Exd binding site affinity and its role in Hox specificity. Our results show that contrary to initial predictions, the putative low-affinity/non-canonical class 2 Dfd–Exd sites function as high-affinity sites. Dfd–Exd complexes bind strongly to these sites with the cell-specific activation of *AP2x* resulting from a balance between the affinity of Dfd–Exd binding sites and the concentration levels of Dfd present throughout the maxillary segment. More importantly, this configuration is crucial for Dfd function in the maxillary segment, as it allows Dfd to control and coordinate the morphogenesis of the different maxillary structures. Moreover, we show that despite these sites being bound by Dfd with high affinity, the Dfd–Exd sites are under tight constraints: optimization of these sites leads to an increase in sensitivity of the *AP-2* maxillary enhancer to lower Dfd concentrations, resulting in the ectopic activation of the enhancer in the maxillary segment and consequent disruption of the Dfd-coordinated development of maxillary structures. Furthermore, although Dfd–Exd and Scr–Exd complexes have been shown to bind optimal class 2 binding sites with similar affinities[9], activation of the *AP-2* enhancer was restricted to the maxillary segment and was not induced in segments controlled by other Hox proteins. In sum, our results demonstrate that Dfd–Exd high-affinity sites play an important role in determining the spatial-temporal activation of a Hox target gene. This finding challenges the view based on posterior Hox protein studies that low-affinity Hox–Exd sites are the rule to ensure Hox protein specificity.

## Results

### *AP-2* is required for the development of maxillary structures
During embryogenesis, *AP-2* is expressed in the brain and in the maxillary segment (Fig. 1a, Supplementary Fig. 1a, b), with the latter expression being dependent on the anterior *Hox* gene *Dfd* (Supplementary Fig. 1c–e)[15–17]. Previous genome-wide profiling of Dfd

chromatin interactions had identified the *AP2x* enhancer as a Dfd-regulated CRM activating the dynamic *AP-2* expression in the maxillary segment (Fig. 1a–e′′′, Supplementary Fig. 1c–e)[15]. *AP2x*-mediated expression is initiated in cells located in a dorsal posterior region and along a midline crossing the maxillary segment from anterior to posterior in stage 10 embryos (Fig. 1c, c′), followed by additional activation in posterior ventral cells by stage 12 (Fig. 1d, d′). Full activation is achieved by stages 13–14, which are maintained until the end of embryogenesis (Fig. 1e, e′). In later embryonic stages, we observed the development of specific structures originating from *AP2x*-expressing cells, which were located in close vicinity to the Distal-less (Dll)-expressing dorsal cirri and which resembled the ventral cirri (Supplementary Fig. 2a, b, d, e), specific structures in the head of the *Drosophila* larvae (Supplementary Fig. 2c)[18]. Although *AP-2* has been shown to be required for proboscis, leg, and central brain development as well as for night sleep and the regulation of feeding behaviour[19–23], its function in the maxillary segment has so far not been studied. Thus, we analysed cuticles of 1st instar larvae carrying the *AP-2*[15] null allele[19]. This analysis revealed that the mutant larvae failed to develop ventral cirri (Supplementary Fig. 2f, f′, g, g′), while the dorsal cirri and the mouth hooks, which arise from a primordium in the anterior-ventral part of the maxillary segment (Supplementary Fig. 2a), developed normally (Supplementary Fig. 2f, f′, g, g′).

Taken together, these results showed that *AP-2*, which is dynamically expressed in the maxillary segment, is required for the formation of the ventral row of cirri in the *Drosophila* larva head and that the *AP2x* enhancer recapitulates the maxillary-specific *AP-2* expression.

### The *AP2x-377* element contains the minimal regulatory information
To study Hox-dependent regulation of *AP-2*, we aimed to identify the minimal information controlling maxillary-specific *AP-2* expression. To this end, we analysed the *AP2x* sequence across different *Drosophila* species and found three highly conserved domains spanning the *AP2x* enhancer (Supplementary Fig. 3). Using this information as reference, we dissected the *AP2x* enhancer by systematically deleting regions and analysing reporter gene expression in transgenic lines (Fig. 2a). Deletion of 353 nucleotides at the 5′ end of the *AP2x* enhancer (*AP2x-592*), which included conserved domain 1, as well as deletion of the non-conserved region at the 3′ end (*AP2x-377*) showed almost identical activity to the *AP-2x* enhancer in the maxillary segment (Figs. 2b–d, 2b′–d′), demonstrating that the *AP2x-377* fragment was sufficient for proper maxillary *AP-2* activation. Removal of the 214 nucleotides 3′ to domain 3 resulted in ectopic activation of the *AP2x-377* fragment in the antennal region at stage 11 (Fig. 2b, d), which was not observed in *AP2x-592* embryos (Fig. 2b, c), but in embryos controlling transgene expression under the control of these 214 bp (AP2x-214, Fig. 2b, e), showing that this region is involved in some repressive function in the antennal region at that stage. Domains 2 and 3 located within the *AP2x-377* are important for gene activation, as deletion of either domain (*AP2x-230* and *AP2x-268*) resulted in the loss of maxillary enhancer activity in earlier stages (Fig. 2b, f, g), while activation was reduced at later stages (Fig. 2b′, f′, g′). Similar to the *AP2x-377* element, the *AP2x-230* and *AP2x-268* elements induced ectopic reporter activity in the antennal segment in stage 11 (Fig. 2b, f, g). Despite their importance, domains 2 and 3 were not sufficient to induce reporter gene activation on their own (*AP2x-152, AP2x-109*) (Fig. 2b, b′, h, h′, i, i′). These results suggested that in addition to the conserved domains, the non-conserved region between these two elements is required for enhancer activity. Indeed, deletion of the non-conserved region from either *AP2x-230* or *AP2x-268* resulted in the loss of activity in the maxillary segment (Fig. 2b, f–i, 2b′, f′–i′). Importantly, the *AP2x* and *AP2x-377* enhancers induced reporter gene expression in identical cells confirming that the *AP2x-377* fragment contains minimal information for precise *AP-2* activation in the maxillary segment (Supplementary

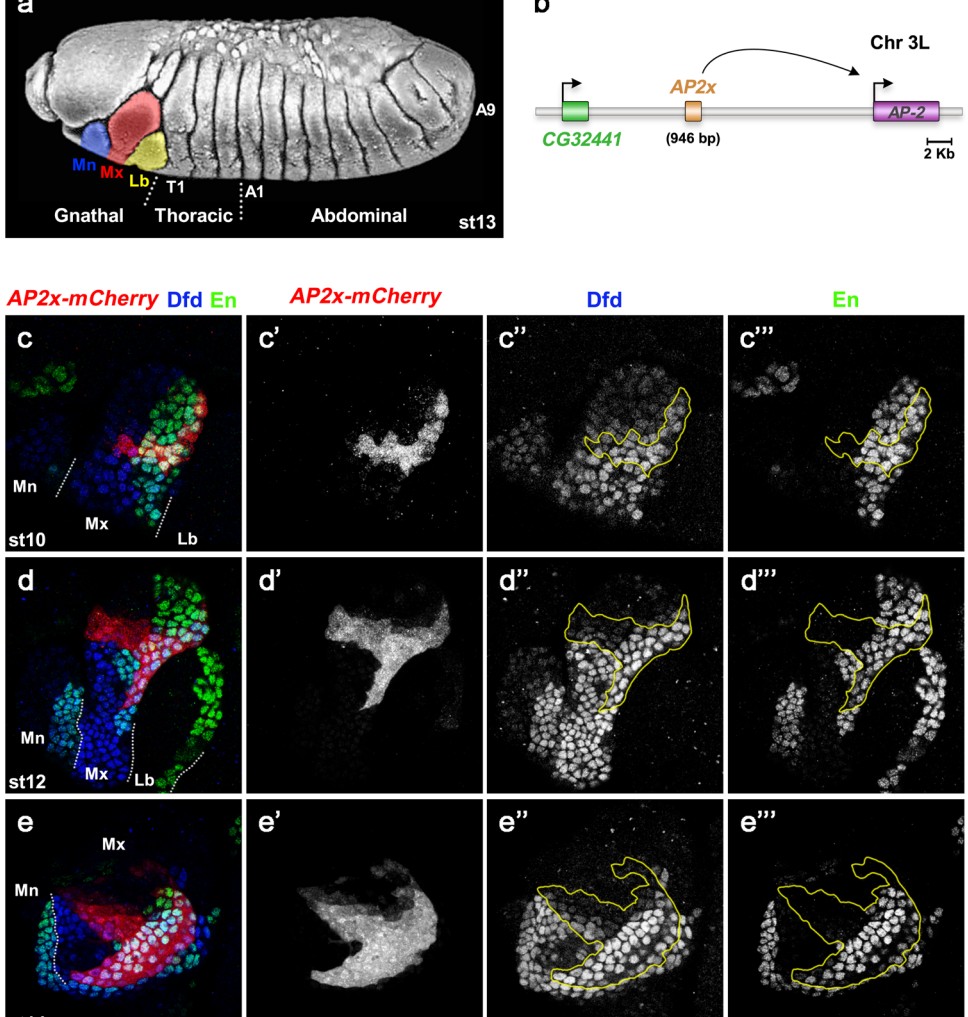

**Fig. 1 | The *AP2x* enhancer is dynamically activated in the maxillary segment during embryogenesis. a** Modified scanning electron microscope image of a stage 13 embryo[37] with the mandibular (Mn, blue) maxillary (Mx, red) and labial (Lb, yellow) segments highlighted. **b** Schematic representation of the *AP-2* locus. The *AP2x* enhancer, located between the *CG32441* and *AP-2* genes, acts specifically on *AP-2*. **c–e'''** Activity of the *AP2x* enhancer during embryogenesis. Embryos carrying the *AP2x-mCherry* reporter stained against mCherry to analyse the activity of *AP2x*. **c–c''** During stage 10, *AP2x* (red in **c** and grey in **c'**) is active in the dorsal posterior maxillary cells and along a midline crossing the segment from anterior to posterior. **d–d'''** In stage 12, posterior ventral cells start to activate *AP2x* (red in **d**, grey in **d'**) with full activation being achieved by stage 14 (red in **e**, grey in **e'**). The maxillary segment was labelled by staining embryos against Dfd (blue in **c**, **d**, **e**, grey in **c''**, **d''**, **e''**), while staining for En labelled the maxillary posterior border cells (green in **c**, **d**, **e**, grey in **c'''**, **d'''**, **e'''**). The yellow lines in **c''**, **c'''**, **d''**, **d'''**, **e''** and **e'''** outline the domain of *AP2x-mCherry* expression. White dashed lines in **c–e** indicates the position of the segmental borders. Mn: Mandibular segment; Mx: Maxillary segment; Lb: Labial segment; T1: 1st Thoracic segment; A1: 1st Abdominal segment; A9: 9th Abdominal segment. See also Supplementary Figs. 1 and 2.

Fig. 4a-b''). Finally, we also tested *AP2x-377* activity in *Dfd^{w21}* null mutants, which resulted in the loss of activity in the ventral posterior border cells of the maxillary segment as well in the medial arm (Supplementary Fig. 4c-d''). Furthermore, as observed for the *AP2x* enhancer, the activity of the *AP2x-377* enhancer in the dorsal posterior cells was independent of Dfd (Supplementary Fig. 1e, 4c-d'').

In sum, these results revealed that the *AP2x-377* fragment contains the minimal information required for Dfd-dependent *AP-2* activation in the maxillary segment.

### Dfd-Exd binds low-affinity binding sites in the *AP2x-377* enhancer

We next performed a comprehensive analysis of the *AP2x-377* sequence to identify all potential Dfd–Exd sites. As we did not identify any canonical class 2 Hox-Exd sites within the *AP2x-377* sequence, we used the No Reads Left Behind (NRLB) algorithm, which identifies the TF binding sequences of diverse affinities[24], on the *AP2x-377*

sequence. This analysis led to the identification of 10 regions predicted to be bound by Dfd-Exd complexes with different, yet relatively low affinities (Fig. 3a–c, Supplementary Fig. 5a, b, Table 1). Comparison to a predicted class 2 canonical Dfd-Exd high-affinity site showed a difference of at least two orders of magnitude in binding affinity (Fig. 3a, Supplementary Fig. 5c). Seven of these sites contained the AYnnAY Hox-Exd sequence and either matched previously described Dfd/Hox-Exd low-affinity sites (sites 1, 2, 6, 7 and 10) or were variations of such sites (sites 5 and 8)[9]. The remaining identified Dfd-Exd sites did not match the AYnnAY Hox-Exd sequence (sites 3/4 and 9) (Fig. 3b) and are thus expected to be also of low-affinity[12]. To test the interaction of Dfd and Exd with these sites, we performed electrophoretic mobility shift assays (EMSAs) using oligonucleotides containing the identified Dfd-Exd sequences (Fig. 4a-c, e, f and Supplementary Fig. 6a-c, e, f). In addition, we used mutated versions of these sequences to test the specific interaction of Dfd and Exd with individual Dfd-Exd sites, as many predicted Dfd-Exd sites overlapped (Supplementary Fig. 5a, b).

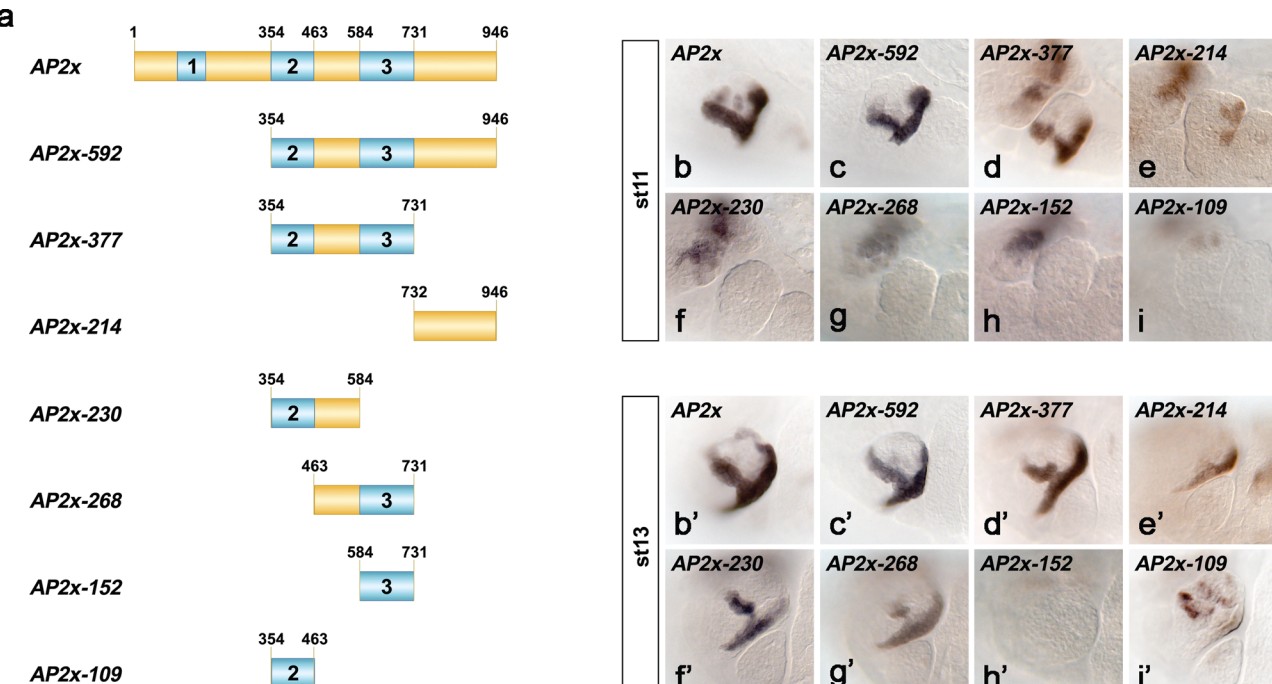

**Fig. 2 | Systematic dissection of the *AP2x* enhancer. a** Schematic representation of elements designed to study the activity of *AP2x*. The three regions in blue (1–3) indicate highly conserved sequences, while the yellow regions are non-conserved. The different fragments were used to generate transgenic mCherry reporter lines. **b-i'** Activity of the different enhancer elements in the maxillary segment during embryonic stages 11 (**b-I**) and 13 (**b'-i'**). The *AP2x-377* element contains the minimal information for driving *AP-2* expression in the maxillary segment, as its enhancer activity is very similar to the full-length *AP2x* enhancer in both developmental stages. See also Supplementary Figs. 3 and 4.

This analysis revealed that the majority of sites were bound by Dfd with binding efficiency increasing in the presence of the Hox cofactors Exd and Hth (Fig. 4b–h, Supplementary Fig. 6b–h). Interestingly, we did not find a clear correlation between predicted relative Dfd–Exd binding affinity and binding site numbers or binding sequences. For example, oligo C located in domain 2, which harbours four Dfd–Exd sites with three of the sites (sites 5, 7, 8) predicted to be among the ones with the highest relative affinities (Fig. 3a, b and Table 1), was found to be bound by Dfd–Exd–Hth less efficiently (Supplementary Fig. 6e, f, h) than oligo A (Supplementary Fig. 6b, c, d, Fig. 4h), which contains only one Dfd–Exd site with a lower predicted relative binding affinity (Fig. 3a, b and Table 1). A detailed analysis of oligo C revealed that only the overlapping sites 5/6 but not sites 7/8 contributed to binding (Supplementary Fig. 6e, f, h). In contrast, oligo D located in domain 3, which harbours two Dfd–Exd sites (sites 9 and 10) with site 9 having a rather high relative affinity (Fig. 3a), required both sites for full binding (Fig. 4b–d). And finally, the binding sites predicted in oligo B (sites 2, and 3/4) act redundantly, as binding was only abolished when all sites were mutated (Supplementary Fig. 6e–g). We also tested an additional sequence present in domain 3, site 11 (GCA**CCTAAT**GAC), which was not retrieved by the NRLB algorithm but showed similarities to other predicted Dfd–Exd sites located in domains 2 and 3 (Fig. 3b, Supplementary Fig. 5b). EMSAs revealed that Dfd–Exd complexes were able to interact with site 11 (Fig. 4e–g) with a similar relative affinity to other sites located in the *AP2x-377* fragment (Fig. 4h).

Overall, with exception of binding sites 5 and 6, all predicted Dfd–Exd sites in *AP2x-377* were bound by Dfd–Exd complexes. While we were able to study some of these sites individually (sites 1, 9, 10 and 11), the substantial overlap of other sites made it difficult to ascertain their individual contribution (sites 2/3/4 and 7/8). Importantly, despite the predicted relatively low affinity, the identified Dfd–Exd sites were bound rather efficiently by Dfd–Exd complexes.

## Dfd–Exd activates the *AP-2* enhancer via Hox–Exd sites in domain 3

To determine the in vivo functionality of the identified Dfd–Exd binding sites present in *AP2x-377*, we generated transgenic lines that carried mutations to disrupt the Dfd–Exd binding sites in the conserved domains 2 and 3 (Fig. 5a). Surprisingly, this analysis revealed that the Dfd–Exd sites present in domain 2 did only minorly contribute to enhancer activity, as expression driven by *AP2x-377-D2mt* was very similar to the one controlled by *AP2x-377* (Fig. 5c-c", d-d"), with the quantification of the activity of both enhancers showing no significant difference in the Dfd-dependent posterior-ventral expression domain (Fig. 5f). This result implied that the *AP2x-377* enhancer activity primarily depends on Dfd–Exd sites present in domain 3, and consistently, mutation of all of these sites (*AP2x-377-D3mt*) resulted in almost complete loss of enhancer activity with only some residual reporter activity observed in a few cells in the medial stripe (Fig. 5c-c", e-e").

We next wanted to provide additional evidence that Dfd–Exd binding sites in domain 3 are of functional relevance for maxillary development. To this end, we performed rescue experiments of ventral cirri in *AP-2*[15] mutants by driving the expression of *UAS-AP2* by GAL4, whose expression was controlled by the different enhancer sequences (*AP2x-377*, *AP2x-377-D2mt*, *AP2x-377-D3mt*). As shown earlier, *AP-2*[15] mutants fail to develop ventral cirri (Supplementary Fig. 2f, f', g, g'). Expressing *AP-2* under the control of the *AP2x-377* sequence resulted in an almost complete rescue of *AP-2*[15] as observed by the development of ventral cirri (Fig. 5g, Supplementary Table 1), showing that this element is required for the control of these maxillary-specific structures. A similar rescue was observed using the *AP2x-377-D2mt* sequence to control GAL4-dependent *AP-2* expression (Fig. 5h, Supplementary Table 1). This is in line with the reporter analysis and confirmed that Dfd–Exd binding sites in domain 2 are not required for the development of these structures. Finally, we did not observe any significant rescue of ventral cirri in *AP-2*[15] embryos when GAL4 was

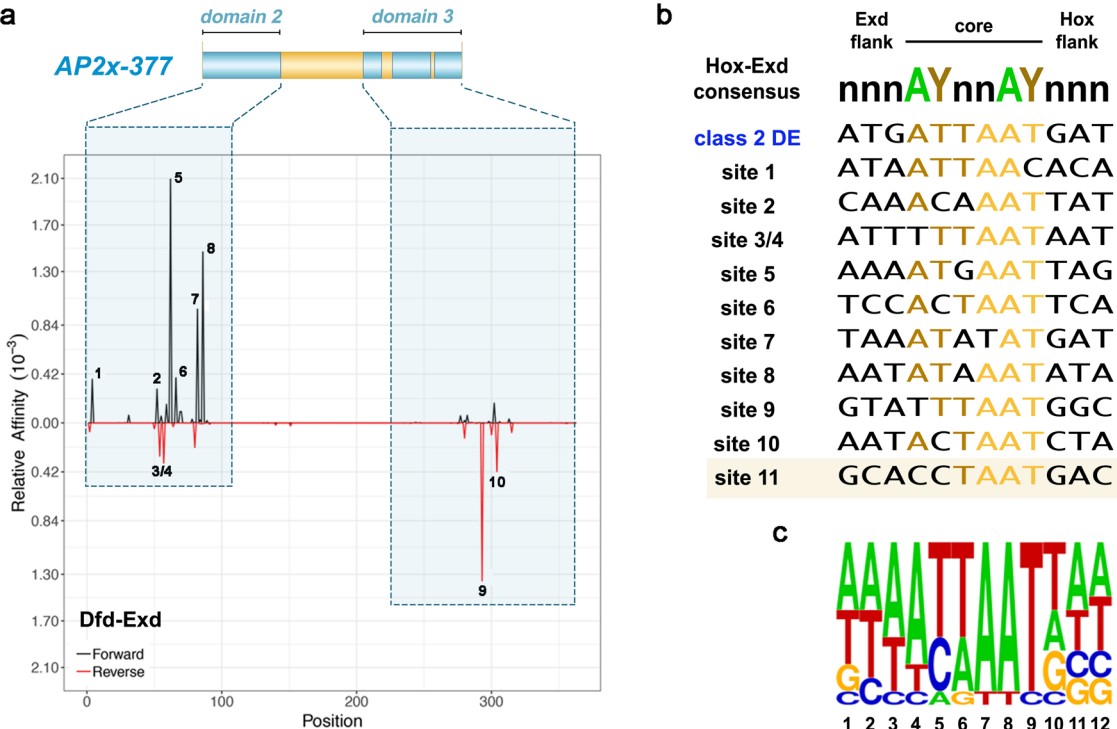

**Fig. 3 | The *AP2x-377* enhancer contains several predicted low-affinity Dfd–Exd binding sites. a** Dfd–Exd binding regions in *AP2x-377* predicted by the *No Reads Left Behind* (NRLB) algorithm. The identified regions are located in the conserved domains 2 and 3. **b** Alignment of an optimal high-affinity class 2 Dfd–Exd binding site (class 2 DE) with the putative Dfd–Exd binding sites present in *AP2x-377*. Sites 1–10 were identified in the Dfd–Exd NRLB predicted regions with sites 3 and 4 showing a considerable overlap. Site 11 (highlighted by a box) was not detected by the NRLB algorithm but due to similarities to known characterized low-affinity binding sites[12]. Nucleotides highlighted in yellow and brown display 80–100% and 60–80% similarity among all sequences aligned, respectively. Non-highlighted nucleotides display <60% similarity among all sequences aligned. **c** *Weblogo* frequency plot of the predicted Dfd–Exd binding sites present in *AP2x-377*. See also Supplementary Fig. 5.

controlled by the *AP2x-377-D3mt* sequence (Fig. 5i, Supplementary Table 1). Of the 50% *AP-2*[15] larvae overexpressing *AP-2* under the control of *AP2x-377-D3mt*, only 28% were able to develop structures that vaguely resembled cirri, therefore highlighting the functional importance of the Dfd–Exd sites present in domain 3 for maxillary-specific *AP-2* expression and ventral cirri development.

In sum, these results showed that Dfd–Exd complexes activate *AP2x-377* primarily through Dfd–Exd sites present in domain 3. As we have shown before that domain 2 is required for enhancer activity (Fig. 2a, d, d′ g, g′), we assume that this domain contains binding sites for additional factors controlling *AP-2* maxillary expression.

### Optimization of Dfd–Exd sites results in enhancer misregulation

It has been shown before that different Hox binding affinities are critical determinants of Hox specificity[12,14]. For example, in the case of the Hox-dependent *sub* enhancer, optimization of Ubx low-affinity sites to high-affinity sites converted the specific regulation of this enhancer by Ubx to one responding also to anterior Hox input[12]. According to the NRLB algorithm, all Dfd sites present in the *AP2x-377* enhancer are of relatively low affinity (Fig. 3a). Thus, we wondered how this enhancer responded to optimization of these sites. To this end, we converted all sites present in domain 2 or 3 to sequences that should be optimally bound by Dfd–Exd complexes by changing the Hox core as well as the flanking sequences to resemble class 2 Hox–Exd sites, as demonstrated here for sites 10 and 11 (Supplementary Figs. 7a, 8a). Oligonucleotides containing the different modifications were used for EMSAs to quantitatively compare the binding affinities of Dfd–Exd complexes to the different binding sites. To this end, we calculated the equilibrium dissociation constants ($K_D$) for the interaction of the Dfd–Exd complexes with the wild-type and optimized version of Dfd–Exd sites 10

and 11, as well as for the interaction with the canonical class 2 Dfd–Exd high-affinity site. We found that optimization of the Dfd–Exd site 10 -**AAT**ACTAATCTA- (oligo D-9mut-10wt $K_D = 16.9$ nM) to -**TTGAT**TAATTA**A**- (oligo D-9mut-10H $K_D = 8.16$ nM) generated a 2.1-fold increase in the binding affinity for Dfd–Exd complexes (Supplementary Fig. 8a–c). Similarly, optimization of the Dfd–Exd site 11 -**GCACC**TAATGAC- (oligo E-wt; $K_D = 16.8$ nM) to -**ATGAT**TAATGAC- (oligo E-11-H; $K_D = 6.7$ nM) generated a 2.5-fold increase in the binding affinity (Supplementary Fig. 7a–c). While Dfd–Exd complexes bind with almost similar affinity both the class 2 Dfd–Exd high-affinity site and the optimized site 10 (Supplementary Figs. 7d, 8c), the binding affinity of Dfd–Exd for the optimized site 11 was 1.4-fold higher than for the canonical class 2 sequence (Supplementary Fig. 7a, c, d). Altogether, these results were surprising because contrary to the NRLB-based predictions they showed that Dfd–Exd sites 10 and 11 in the *AP2x-377* enhancer are of high affinity and converting them to an optimal class 2 Hox–Exd sequences resulted only in a modest 2–2.5-fold increase in the binding affinity for Dfd–Exd complexes (Supplementary Figs. 7, 8).

In the next step, we tested the binding of Dfd protein to the different *AP2x-377* enhancer versions in vivo. To this end, we performed chromatin immunoprecipitation (ChIP) experiments using a Dfd antibody on chromatin retrieved from embryos containing the wild-type version of the *AP2x-377* transgene (*AP2x-377*), the mutated version of the enhancer (*AP2x-377-D3mt*) and the version containing the Dfd high-affinity sites (*AP2x-377-D23H*). To specifically analyse Dfd binding to the enhancer transgene and not the endogenous *AP-2* enhancer, qPCR was performed on precipitated DNA using primers that were located in the *AP2x-377* enhancer transgene and in the vector backbone. We found enrichment of Dfd binding to both the *AP2x-377* wildtype version and when the Dfd binding sites in the *AP2x-377* enhancer were

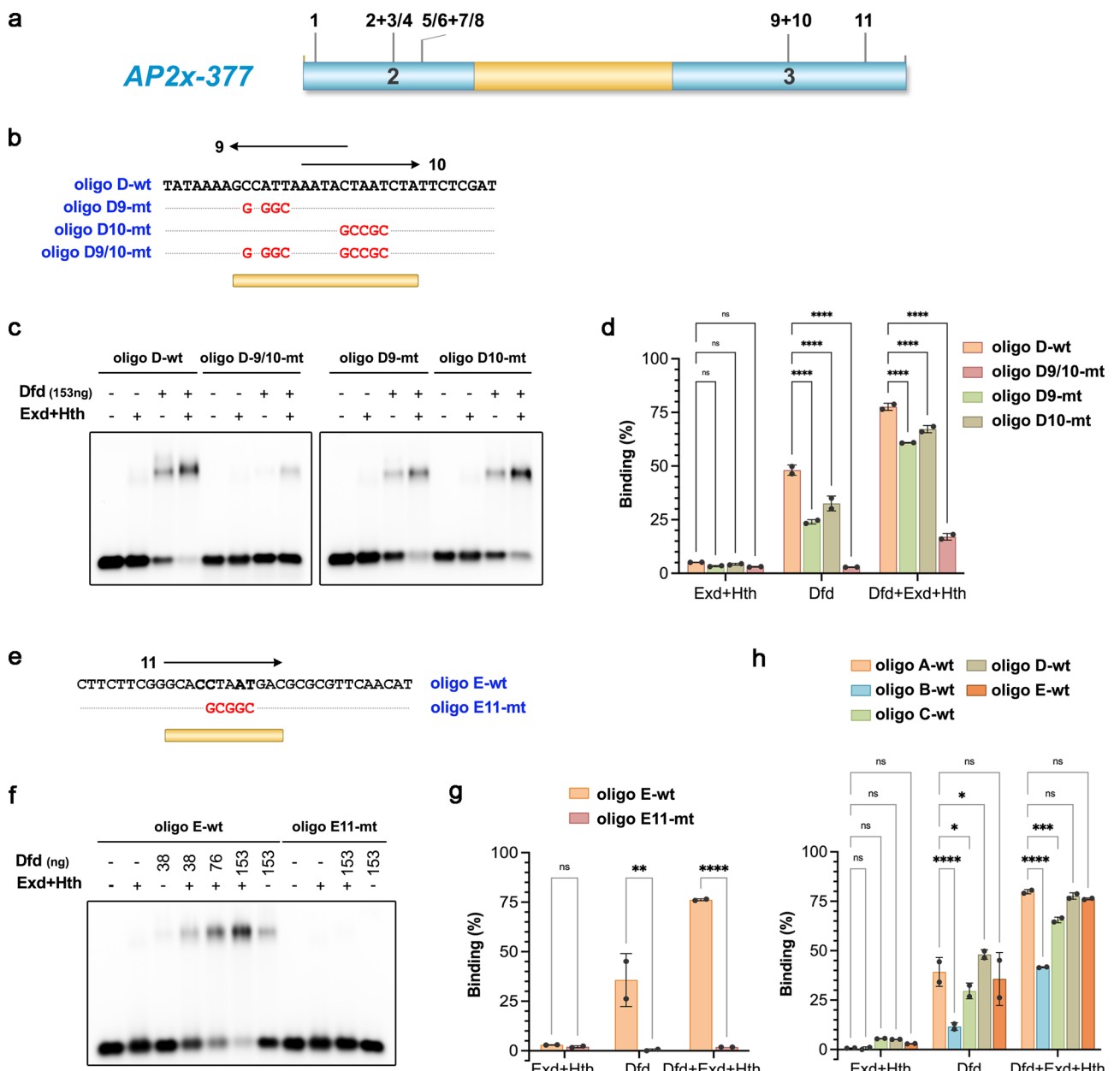

**Fig. 4 | Dfd–Exd complexes bind in vitro to predicted Dfd–Exd binding sites present in the *AP2x-377* enhancer. a** Schematic representation of the *AP2x-377* enhancer with the predicted Dfd–Exd binding sites indicated. **b, e** Oligonucleotides designed to overlap *AP2x-377* regions containing predicted Dfd–Exd binding sites (sites 9 and 10 in **b** and site 11 in **e**); as a control, oligonucleotides were designed with mutations to disrupt the predicted Dfd–Exd sites (**b, e**). **c, f** EMSAs using the oligonucleotides shown in **b** and **e** with Dfd in the presence of Exd and Hth. **d, g, h** The quantification of Dfd–Exd relative binding affinities to the predicted

Dfd–Exd binding sites present in *AP2x-377* was determined by calculating the ratio of the bound probe to unbound free probe (analysed EMSAs *n* = 2). The plots indicate the mean and the corresponding standard deviation values. Yellow bars in **b, e** highlight the Dfd–Exd binding sites in the respective oligos. ns: non-significant; *p*-value = 0.0239 (in **h**, oligo A-wt vs. oligo C-wt) and *p*-value = 0.0439 (in **h** oligo A-wt vs. oligo D-wt); **p*-value = 0.0020; ***p*-value = 0.0004; ****p*-value < 0.0001. See also Supplementary Fig. 6. Source files are provided in "Source-Data-File_values".

replaced with high-affinity versions (Supplementary Fig. 8d). Notably, the ChIP signals suggest increased binding of the Dfd protein to the optimized *AP2x-377-D23H* enhancer. However, we can also not exclude the possibility that differences in ChIP signal are wholly or partly caused by differences in the number of maxillary cells responding to Dfd in the *AP2x-377-D23H* enhancer context.

Finally, we determined the effect of this increase in in vitro Dfd–Exd affinity on in vivo enhancer activity. Transgenic embryos carrying the *AP2x-377* enhancer with all Dfd–Exd sites converted to high-affinity Dfd–Exd sequences (*AP2x-377-D23H*) expressed the reporter gene in ectopic locations in the maxillary segment (Fig. 6a, b, c-c" and f-f"). In contrast to the restricted activation of *AP2x-377* in cells

expressing high levels of Dfd in the posterior border of the maxillary segment (Fig. 6c-c"), *AP2x-377-D23H* was ectopically activated in cells anterior and ventral to this region (Fig. 6f-f"). Intriguingly, these cells expressed low levels of Dfd (Fig. 6f"), while in posterior cells with high Dfd expression levels (Fig. 6f"), the reporter gene was not increased in the presence of high-affinity sites (Fig. 6i). Consistent with previous results, changes in the activation of the enhancer were mediated by domain 3 only, as the conversion of the Dfd–Exd binding sites in this domain resulted in a similar ectopic reporter gene activation (*AP2x-377-D3H*) in anterior and ventral cells of the maxillary segment when compared to *AP2x-377-D23H* embryos (Fig. 6a, d, d', f, f'). In contrast, the presence of high-affinity sites in domain 2 alone had no effect (Fig.

**Table 1 | NRLB analysis of Dfd-Exd binding regions in *AP2x-377***

| Site | Affinity | Position* | Sequence |
|------|----------|-----------|----------|
| 1 | 0.0003754535 | 355 | CGATAATTAACACACAAG |
| 2 | 0.0002919404 | 403 | ACCAAACAAATTATTAAA |
| 3 | 0.0002855378 | −405 | CAAACAAATTATTAAAAA |
| 4 | 0.0003443779 | −408 | ACAAATTATTAAAAATGA |
| 5 | 0.0020940541 | 413 | TTATTAAAAATGAATTAG |
| 6 | 0.0003880488 | 417 | TAAAAATGAATTAGTGGA |
| 7 | 0.0009773018 | 433 | GAAATATAAATATATGAT |
| 8 | 0.0014693645 | 437 | TATAAATATATGATTTTG |
| 9 | 0.0013565491 | −644 | AAAAGCCATTAAATACTA |
| 10 | 0.0004173395 | −655 | AATACTAATCTATTCTCG |

*(the position refers to the location in *AP2x*).

6a, c, c', e, e'). In addition, high-affinity sites in domain 2 failed to rescue the loss of enhancer activity observed when Dfd–Exd sites in domain 3 were mutated (Fig. 6a, e, e', h, h', *AP2x-377-D2H-D3mt*). And conversely, ectopic activation by *AP2x-377-D3H* was unchanged when Dfd–Exd sites in domain 2 were mutated (Fig. 6a, d, d', g, g'). These results showed that domain 2 Dfd–Exd binding sites are non-functional even when converted to high-affinity sites in this tissue context. An increase in binding affinity of Dfd–Exd not only affected cell specificity but also temporal regulation of *AP-2* (Supplementary Fig. 9). In contrast to *AP2x-377*, which was activated during early stage 11 in specific domains of the maxillary segment (Supplementary Fig. 9c-c'''), *AP2x-377-D23H* was active throughout the maxillary segment as early as stage 10 (Supplementary Fig. 9a-b''', d-d''').

Despite the spatial and temporal mis-regulation in the maxillary segment, none of the changes were sufficient to drive gene expression in other segments, nor in the segment located posterior to the maxillary segment, the labial segment (Supplementary Fig. 10a-a'' and c-c''), which is under the control of the Hox TF Sex combs reduced (Scr)[25]. This was surprising since Dfd– and Scr–Exd complexes have been shown to bind class 2 Hox–Exd sites with comparable high affinities[9]. Consistent with our in vivo data, Scr–Exd complexes bound the *AP2x-377* and *AP2x-377-D3H* enhancers with strongly reduced affinities in comparison to Dfd–Exd complexes, as demonstrated for site 11 (Supplementary Fig. 11a-c). Intriguingly, Scr–Exd complexes displayed a 7-fold lower affinity for site 11 compared to Dfd-Exd (Supplementary Fig. 7b and 11b). In striking contrast to Dfd–Exd complexes, converting site 11 to an optimal class 2 Hox-Exd binding site further decreased the affinity of Scr-Exd complexes by 4.6-fold (Supplementary Fig. 11b, c).

In sum, these results showed that in vitro, and contrary to the initial predictions based on the NRLB algorithm, Dfd-Exd sites present in *AP2x-377* are of relatively high affinity and optimization of these sites to canonical class 2 sites further increased their affinity to Dfd-Exd complexes. Moreover, although being high-affinity sites, the Dfd-Exd sites in *AP2x-377* are tightly regulated: increasing the affinity of these sites induced a spatial and temporal mis-regulation of the enhancer in the maxillary segment, in particular in cells with lower Dfd expression levels, while enhancer activity was unchanged in cells which already expressed high Dfd levels. Furthermore, although Dfd–Exd and Scr–Exd are able to bind to the same sequences[9], Scr–Exd complexes exhibited a strongly reduced affinity for the Hox–Exd sites present in *AP2x-377* even after their conversion to class 2 binding sites.

**Dfd–Exd sites in *AP-2* enhancer are sensitive to Dfd levels**

It has been shown recently that the Hox dosage plays a very important role in morphogenesis[26]. Thus, we tested the hypothesis that specificity in regional Hox target gene activation depends on differences in Hox–Exd binding affinities and Hox protein levels. To this end, we first

asked whether the ectopic activity of the *AP2x-377-D23H* enhancer in anterior-ventral maxillary cells driven by Dfd-Exd high-affinity sites (Fig. 6f, f') could be changed when Dfd levels are reduced in this region. To this end, we expressed an UAS-*Dfd*^RNAi^ transgene in anterior-ventral maxillary cells by means of the *patched (ptc)*-GAL4 driver[27] (Fig. 7f), which is active specifically in these cells in the maxillary segment (Fig. 7j-j''). We found that decreasing the levels of Dfd by 2-fold in anterior-ventral maxillary cells caused a 2-3-fold reduction in the *AP2x-377-D23H* enhancer activity (Fig. 7g-i'', l), with the overall activation of the enhancer resembling the expression driven by *AP2x-377* (Fig. 6c, c'). Vice versa, we also tested whether an increase of Dfd levels in anterior-ventral maxillary cells could activate the *AP2x-377* enhancer in these cells. To this end, we expressed an UAS-*Dfd* transgene in maxillary cells by means of the *AP2x-377-D23H*-GAL4 driver (Fig. 7a). This driver is under the control of the *AP2x* enhancer harbouring Dfd-Exd high-affinity sites, thus it activates transgene expression in all *AP-2* expressing cells and induces ectopic activation in anterior-ventral maxillary cells (Fig. 7e-e''). Dfd expression was increased in all cells targeted by the *AP2x-377-D23H*-GAL4 driver (Fig. 7b'', c'', d'', k), resulting in an increase of *AP2x-377* enhancer activity in all cells normally expressing *AP-2* as well as in ectopic enhancer activity in anterior maxillary cells (Fig. 7b', c', d', k). These results showed that the *AP-2* enhancer responds to different Dfd levels in the maxillary segment and is normally unresponsive in anterior maxillary cells due to the low Dfd levels in this region and not (or only to a minor extent) due to chromatin accessibility. To provide further support for our hypothesis that *AP-2* enhancers harbouring Dfd–Exd sites of different affinities are activated by different Dfd levels, we analysed the response of the *AP2x-377* and *AP2x-377-D23H* enhancers in embryos ectopically expressing Dfd in regions outside of the maxillary segment. When Dfd was ubiquitously mis-expressed, *AP2x-377* was activated in anterior cephalic regions, specifically in the antennal segment (Supplementary Fig. 10a-a'' and b-b''). The same was observed when Dfd was over-expressed in *AP2x-377-D23H* embryos (Supplementary Fig. 10c-c'', d-d''). However, *AP2x-377-D23H* was also active in more posterior segment, from the labial to the third thoracic segment (Supplementary Fig. 10c-c'', d-d''), indicating that the class 2 Dfd-Exd high-affinity sites in *AP2x-377-D23H* are more sensitive to Dfd levels.

Altogether, these results showed that the Dfd-Exd sites present in the *AP-2* enhancer are sensitive to the expression levels of Dfd and support the view that the domain-specific activation of *AP-2* in the maxillary segment depends on an interplay between Hox-Exd affinity and Hox dosage.

**Optimization of Dfd–Exd sites impairs maxillary development**

*AP-2* expression in the maxillary segment, which we found to depend on a tight interplay between Dfd protein levels and the nature of Dfd–Exd binding sites, is critical for the development of ventral cirri (Supplementary Fig. 2f, f', g, g'). We next asked how ectopic *AP-2* activation mediated by class 2 canonical Dfd–Exd high-affinity binding sites affected maxillary morphogenesis. This was particularly interesting, as the optimization of the Dfd–Exd binding sites in the *AP-2x-377* element resulted in its activation in several domains overlapping the primordia of other Dfd-regulated maxillary structures, in particular, the dorsal cirri and mouth hook primordia (Supplementary Fig. 2a)[28]. To address this question, we expressed the *UAS-AP2* transgene in *AP-2*[15] mutants using the *AP2x-377*-GAL4, *AP2x-377-D2H*-GAL4 and *AP2x-377-D23H*-GAL4 drivers. As a read-out, we focused our analysis on the development of dorsal cirri and mouth hooks. Wild-type larvae possess 9-10 dorsal cirri (Supplementary Fig. 2c). However, *AP-2*[15] mutants display a slight reduction in the number of these structures, with an average of 8 cirri (Fig. 8h). Although *AP-2* is not expressed in the dorsal cirri cells, the primordia of both dorsal and ventral cirri are adjacent to one another (Supplementary Fig. 2). Thus, the failure to develop ventral cirri may disturb the development of the dorsal cirri in *AP-2*[15] mutants. However,

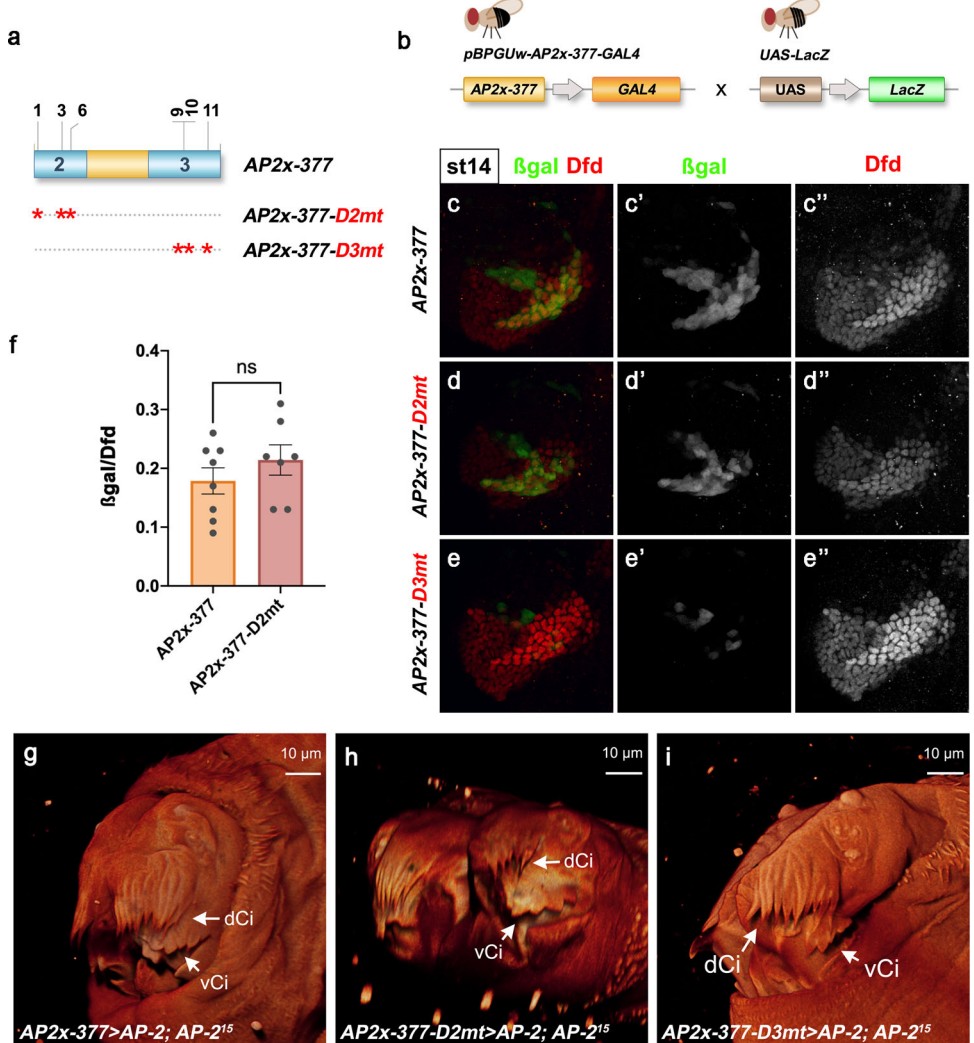

**Fig. 5 | Dfd−Exd activates *AP2x-377* via binding sites in domain 3. a** Schematic representation of mutations inserted in domains 2 and 3 of *AP2x-377* to disrupt the Dfd−Exd binding sites (indicated by an asterisk) identified by EMSAs. Transgenes were generated to express GAL4 under the control of the different mutated *AP2x-377* enhancers. **b** The activity of the mutated *AP2x-377* enhancers was determined by crossing flies carrying the different GAL4 transgenes with *UAS-LacZ* flies. **c−e"** *AP2x-377 > LacZ* (**c−c"**), *AP2x-377-D2mt > LacZ* (**d−d"**) and *AP2x-377-D3mt > LacZ* (**e−e"**) stage 14 embryos were collected and stained for β−galactosidase (green in **c**, **d**, **e**, grey in **c'**, **d'**, **e'**) to determine the activity of the different enhancers in the maxillary segment. Dfd staining labels the maxillary segment (red in **c**, **d**, **e**, grey in **c"**, **d"** and **e"**). **f** Quantification of *AP2x-377* (*n* = 8) and *AP2x-377-D2mt* (*n* = 7) enhancer activity in the posterior border cells of the maxillary segment; *n*: number of embryos analysed. The activity was determined by measuring the β−galactosidase/Dfd fluorescence ratio in the posterior-ventral border cells of the maxillary segment. The plotted values indicate the mean and the corresponding standard error of the mean of the activity of each enhancer. Statistical relevance was tested with the one-way ANOVA test. ns: non-significant. **g−i** Analysis of the ventral cirri in 1st instar *AP-2^15* homozygous larvae cuticles expressing *AP-2* under the control of *AP2x-377-GAL4* (**g**), *AP2x-377-D2mt-GAL4* (**h**) and *AP2x-377-D3mt-GAL4* (**i**). dCi dorsal cirri, vCi ventral cirri. Source files are provided in "Source-Data-File_values".

we did not observe changes in the number of dorsal cirri in *AP2x-377 > AP2* embryos when compared to *AP-2^15* mutants (Fig. 8b, h, Supplementary Table 2). Thus, we assume this reduction in dorsal cirri to be an indirect effect that cannot be rescued by providing normal *AP-2* expression in neighbouring cells. A similar result was obtained when driving *AP-2* with *AP2x-377-D2H*-GAL4 (Fig. 8f, h, Supplementary Table 2), highlighting again the non-functionality of Dfd-Exd sites in domain 2 of the *AP-2* enhancer. In contrast, expression of *AP-2* under the control of the *AP2x-377-D3H* enhancer resulted in a strong reduction in the number of dorsal cirri, with average of five dorsal cirri developed in these animals (Fig. 8g, h, Supplementary Table 2). A similar impairment of maxillary structures was also observed for the mouth hooks. *AP2x-377 > AP-2* larvae displayed a normal set of mouth hooks (Fig. 8i, l). However, embryos expressing *AP-2* under the control of the *AP2x-377-D3H* had severe mouth hook defects (Fig. 8j). Of the 50% of larvae overexpressing *AP-2*, 34% displayed an abnormal set of mouth hooks with the observed phenotypes ranging from malformation in the mouth

hook set (26%) to partial or total loss of these structures (8%) (Fig. 8m). As a control, we overexpressed *AP-2* using the *AP2x-377-D2H*-GAL4 driver, which resulted in the development of a normal set of mouth hooks (Fig. 8k, n).

Taken together, these results showed that modest changes in Dfd-Exd binding affinity in the *AP2x-377* element result in the loss of domain specificity in the maxillary segment, which in turn leads to an impairment of the morphogenesis of Dfd-regulated maxillary structures. Therefore, these results demonstrate that Dfd−Exd binding sites present in the *AP2x-377* element are not only required for the normal development of the ventral cirri, but their fine-tuned affinity is essential for Dfd to coordinate the development of the other maxillary structures, such as the dorsal cirri and the mouth hooks.

## Discussion
Dfd−Exd complexes bind a wide range of sequences with different affinities in vitro[9]. Of these, Dfd−Exd complexes exhibit a higher

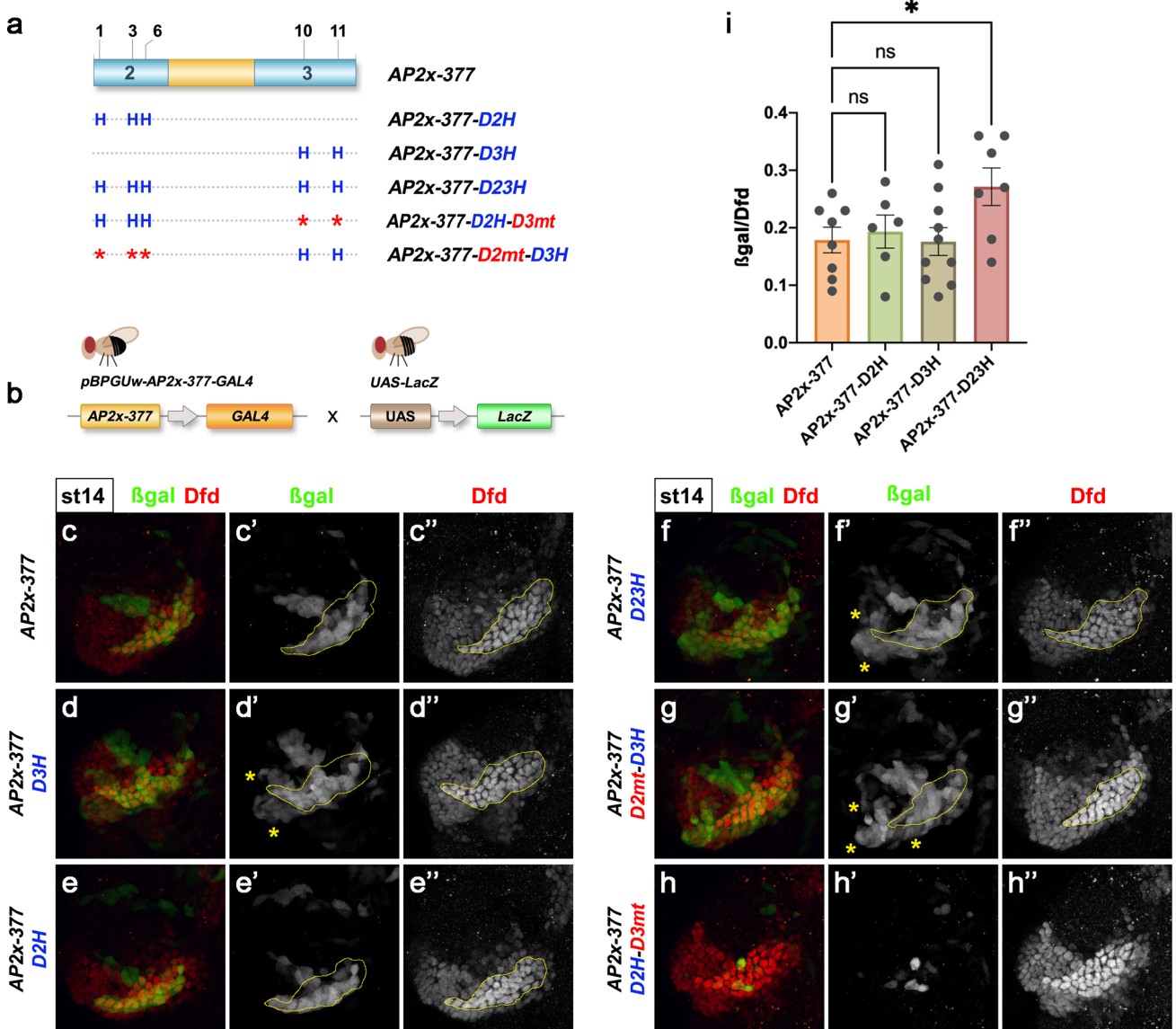

**Fig. 6 | Optimization of Dfd–Exd binding sites results in spatial mis-regulation of the *AP2x-377* enhancer. a** Schematic representation of mutations inserted in the *AP2x-377* to either disrupt (indicated by an asterisk) or optimize (indicated by an H) Dfd–Exd binding sites present in conserved domains 2 and 3. The mutated enhancers were used to generate transgenes driving the expression of GAL4. **b** The activity of the mutated *AP2x-377* enhancers was determined by crossing flies carrying the different GAL4 transgenes with *UAS-LacZ* flies. **c–h"** *AP2x-377 > LacZ* (**c-c"**), *AP2x-377-D3H > LacZ* (**d-d"**), *AP2x-377-D2H > LacZ* (**e-e"**), *AP2x-377-D23H > LacZ* (**f-f"**), *AP2x-377-D2mt-D3H > LacZ* (**g-g"**) and *AP2x-377-D2H-D3mt > LacZ* (**h-h"**) stage 14 embryos were collected and stained for β–galacto-sidase (green in **c, d, e, f, g, h**, grey in **c', d', e', f', g', h'**) to determine the activity of the different enhancers in the maxillary segment. Dfd staining labelled the

maxillary segment (red in **c, d, e, f, g, h**, grey in **c", d", e", f", g", h"**). The yellow lines in (**c', c", d', d", e', e", f' a, f", g'** and **g"**) outline the domain of high Dfd expression in the maxillary segment. The asterisks in **d', f'** and **g'** indicate the ectopic expression of the reporter gene. **i** Quantification of the activity of *AP2x-377* (*n* = 8), *AP2x-377-D2H* (*n* = 6), *AP2x-377-D3H* (*n* = 10) and *AP2x-377-D23H* (n = 7) enhancers; *n*: number of embryos analysed. The activity was determined by measuring the β-galactosidase/Dfd fluorescence ratio in the posterior-ventral border cells of the maxillary segment. The plotted values indicate the mean and the corresponding standard error of the mean of the activity of each enhancer. Statistical relevance was tested with the one-way ANOVA test. *\*p*-value = 0.0320, ns: non-significant. See also Supplementary Figs. 7, 8 and 9. Source files are provided in "Source-Data-File_values".

---

affinity for nTGATTAATnnn core sequences that constitute the class 2 Hox–Exd binding sites. Our analysis of the *AP2x-377* element identified several highly conserved Dfd–Exd binding sites. However, none of these sites matched the high-affinity core sequence of class 2 sites, with some not matching the AYnnAY consensus of Hox–Exd binding sites. In fact, based on their sequences and according to the NRLB algorithm used to identify these sites, all Dfd–Exd binding sites were predicted to be of low affinity. However, our experiments showed that all sites were bound by Dfd–Exd complexes with similar relative affi-nities. Moreover, when compared to class 2 Dfd–Exd high-affinity sites, the identified sites behaved as high-affinity binding sites. As

demonstrated for sites 10 and 11 of domain 3 of *AP2x-377*, Dfd–Exd complexes bound these sites with high-affinity in vitro. The equili-brium dissociation constants ($K_D$) for a canonical class 2 Dfd–Exd site and *AP2x-377* sites 10 and 11 showed that Dfd–Exd complexes exhib-ited only a 1.8-fold higher affinity for the canonical sequences. Con-sistent with these values, the optimization of binding sites 10 and 11 to class 2 high-affinity core sequences resulted in a 2 and 2.5-fold increase in binding affinity, respectively. These results were surprising for two reasons: first, they showed that Dfd–Exd complexes bind with high affinity to a wider range of sequences than initially predicted, and secondly, they suggested that additional flanking sequences might be

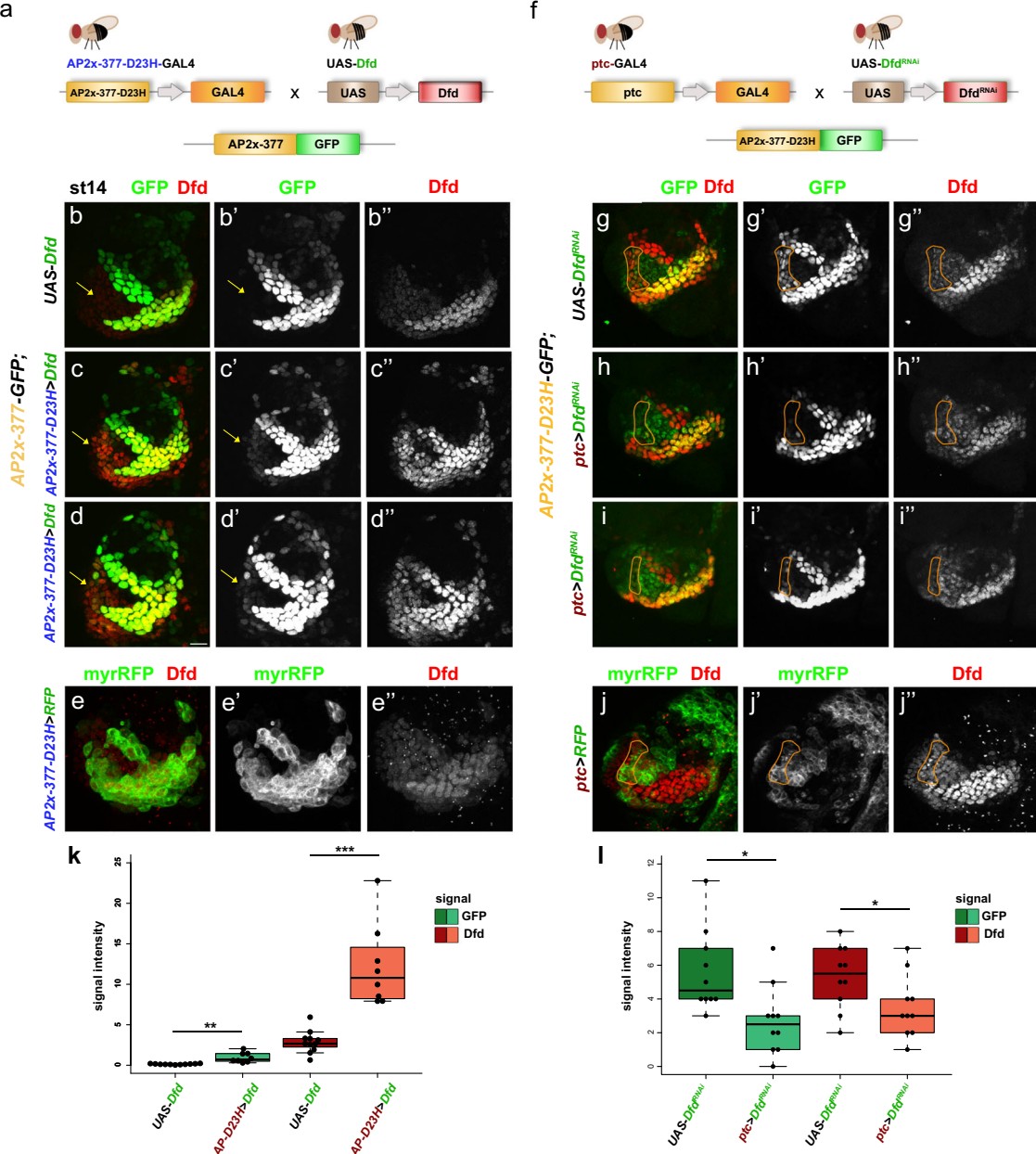

**Fig. 7 | Dfd-Exd sites in the *AP-2* enhancers respond to changes in Dfd levels.**
**a** Schematic outline of the experimental approach used to determine the effects of increased Dfd levels in the activity of *AP2x-377-Luc:GFP (AP2x-377-GFP)*. To increase Dfd levels in the anterior part of the maxillary segment, the UAS-*Dfd* transgene was activated by means of the *AP2x-377-D23H*-GAL4 driver and the activity of *AP2x-377-GFP* was quantified. **b−d"** *AP2x-377-GFP; + > Dfd* (**b-b"**) and *AP2x-377-GFP; AP2x-377-D23H > Dfd* (**c-d"**) stage 14 embryos were stained for GFP (green in **b**, **c**, **d**; grey in **b'**, **c'**, **d'**) and Dfd (red in **b**, **c**, **d**; grey in **b"**, **c"**, **d"**). Yellow arrows mark maxillary anterior-ventral cells. Scale bar: 10 μm. (**e-e"**) *AP2x-377-D23H > myrRFP* embryos were stained for myrRFP (green in **e**; grey in **e'**) and Dfd (red in **e**; grey **e"**), highlighting the myrRFP expression induced by the *AP2x-377-D23H*-GAL4 driver. Stronger GFP expression in posterior and medial maxillary cells is not due to over-exposure of the images (**c'**, **d'**), as all images were taken at same settings, but due to increased Dfd expression resulting in stronger GFP induction. **f** Schematic outline of the experimental approach used to determine the effects of reducing Dfd levels in the activity of *AP2x-377-D23H-Luc:GFP (AP2x-*

*377-D23H-GFP)*. To reduce Dfd levels in anterior maxillary cells, the UAS-*Dfd*<sup>RNAi</sup> transgene was activated by means of the *ptc*-GAL4 driver and the activity of *AP2x-377-D23H-GFP* was quantified. **g-i"** *AP2x-377-D23H-GFP; + > Dfd*<sup>RNAi</sup> (**g-g"**) and *AP2x-377-D23H-GFP; ptc > Dfd*<sup>RNAi</sup> (**h-i"**) stage 14 embryos were stained for GFP (green in **g**, **h**, **i**; grey in **g'**, **h'**, **i'**) and Dfd (red in **g**, **h**, **i**; grey in **g"**, **h"**, **i"**). The orange lines highlight anterior-ventral maxillary cells in which the *ptc*-GAL4 driver is active (**j**, **j'**). **j-j"** *ptc > myrRFP* stage 14 embryos were stained for myrRFP (green in **j**; grey in **j'**) and Dfd (red in **j**; grey **j"**), highlighting the myrRFP expression induced by the *ptc*-GAL4 driver. **k**, **l** Quantification of the expression levels of the GFP reporter and Dfd in the anterior-ventral region of the maxillary segment. In **l**, the area encircled by the orange line in **g-g"**, **h-h"** and **i-i"** was quantified. The plotted box plot represents the collected data shown in dots. Statistical relevance was tested with the two-sided *t*-test: (**k**) **p*-values = 0.00037, ***p*-values = 2.08E−5, (**l**) *p*-value = 0.0103, Dfd: *p*-value = 0.044 (tested embryos *n* > 8). See also Supplementary Figs. 10 and 11. Source files are provided in "Source-Data-File_values".

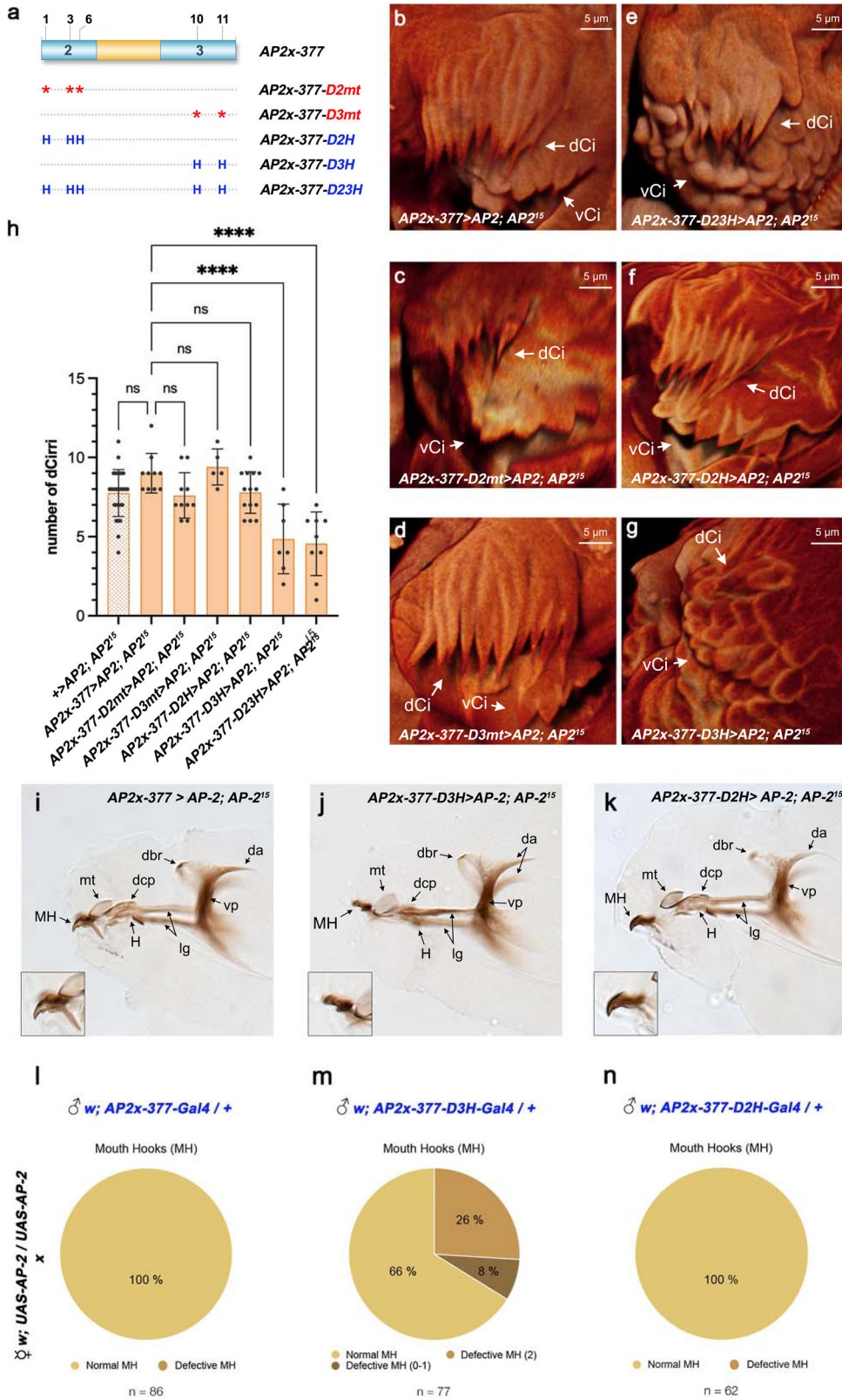

important in determining the overall affinity of Dfd–Exd binding. Although the NRLB has correctly predicted the existence of Dfd–Exd binding sites in *AP2x-377*, it failed to provide a relative affinity that accurately reflected the experimentally determined equilibrium dissociation constant. The NRLB algorithm has been shown to efficiently determine binding sites for several TFs including Ubx-Exd low-affinity sites present in the *E3N* and *7H shavenbaby* enhancers[24]. A possible explanation for this discrepancy could be the SELEX data used to develop the algorithm. In these experiments, oligonucleotides were used that harboured a 12-nucleotide central core and all possible nucleotides permutations of this core, as well as invariant flanking sequences shared by all oligonucleotides. These flanking sequences might account for the discrepancy in binding affinities of the SELEX oligonucleotides and Hox–Exd sequences naturally occurring in

**Fig. 8 | Optimization of the Dfd-Exd binding sites in the *AP2x-377* enhancer impairs the development of dorsal cirri and mouth hooks. a** Schematic representation of mutations inserted in the *AP2x-377* to either disrupt (indicated by an asterisk) or optimize (indicated by an H) Dfd–Exd binding sites present in conserved domains 2 and 3. The modified enhancers were used to generate transgenes driving the expression of GAL4. **b–g** Dorsal cirri (dCi) development in 1st instar *AP-2*[15] homozygous larvae cuticles expressing *UAS-AP-2* under the control of the *AP2x-377-GAL4* (**b**), *AP2x-377-D2mt-GAL4* (**c**), *AP2x-377-D3mt-GAL4* (**d**), *AP2x-377-D23H-GAL4* (**e**), *AP2x-377-D2H-GAL4* (**f**) and *AP2x-377-D3H-GAL4* (**g**). **h** Quantification of the dorsal cirri present in the different *AP-2*[15] rescued larvae: +>*AP2; AP2*[15] (*n* = 28), *AP2x-377 > AP2; AP2*[15] (*n* = 10), *AP2x-377-D2mt > AP2; AP2*[15] (*n* = 10), *AP2x-377-D3mt > AP2; AP2*[15] (*n* = 5), *AP2x-377-D2H > AP2; AP2*[15] (*n* = 14), *AP2x-377-D3H > AP2; AP2*[15] (*n* = 7) and

*AP2x-377-D23H > AP2; AP2*[15] (*n* = 9); *n*: number of larvae analysed. The plotted values indicate the mean and the corresponding standard error of the mean. Statistical relevance was tested with the one-way ANOVA test. ****$p$-value < 0.0001, ns: non-significant. **i–k** Cephalopharyngeal apparatus of *AP-2*[15] homozygous larvae expressing *UAS-AP-2* under the control of *AP2x-377-GAL4* (**i**), *AP2x-377-D3H-GAL4* (**j**) and *AP2x-377-D2H-GAL4* (**k**). The insets highlight the mouth hooks present in these larvae. **l–n** Quantification of abnormal mouth hooks present in larvae expressing *UAS-AP-2* under the control of *AP2x-377-GAL4* (**l**) (*n* = 86), *AP2x-377-D3H-GAL4* (**m**) (*n* = 77) and *AP2x-377-D2H-GAL4* (**n**) (*n* = 62); *n*: number of larvae analysed. dCi dorsal cirri, vCi ventral cirri, MH mouth hooks, mt median tooth, H H-piece, dcp dorsal clasp, lg lateralgräten, dbr dorsal bridge, vp vertical plate, da dorsal arm. Source files are provided in "Source-Data-File_values".

enhancers throughout the genome. Thus, even though the NRLB algorithm faithfully predicts Hox–Exd sites, the corresponding affinities of these sites should be validated by independent experiments.

Although in vitro Dfd–Exd complexes bound several binding sequences present in domains 2 and 3 of *AP2x-377*, the mutation of these sites showed that in vivo only binding sites in domain 3 were required to activate the *AP2x-377* element. Converting the Dfd–Exd binding sites in domain 2 into optimal class 2 high-affinity sites produced no changes in enhancer activity: it did not result in the ectopic activation of the maxillary enhancer neither was it sufficient to rescue the inactivation of Dfd–Exd binding sites in domain 3. However, the introduction of optimal Dfd–Exd binding sites in domain 3 led to a loss of domain specificity. This indicates that, in addition to the type of Dfd–Exd sites, the context is important for the domain-specific activity of the *AP2x-377* element.

Our analysis of the Dfd–Exd sites present in *AP2x-377* also indicates that the establishment of Dfd-specific interactions with ubiquitous cofactors is likely to be required for the activation in the maxillary segment. Misexpression of Dfd throughout the embryonic ectoderm activated *AP2x-377-D23H* in the posterior cells of the labial as well as the thoracic segments, indicating that the inputs required to activate this enhancer are present in these cells. These findings are in alignment with recent studies showing that the Hox TF Ubx relies on the specific interaction with ubiquitous cofactors rather than with cell-specific ones[3]. However, although Dfd–Exd and Scr–Exd complexes have similar affinities to class 2 high-affinity sequences[9], *AP-2* is not activated in the labial segment. One possible reason could be the differential affinity of Dfd and Scr to the Hox–Exd binding sites present in *AP-2*. The determination of the equilibrium dissociation constants of Dfd–Exd and Scr–Exd complexes revealed that Dfd–Exd complexes exhibit a 7-fold higher affinity than Scr–Exd to the Hox–Exd sites present in *AP2x-377*. Moreover, while optimization of *AP2x-377* site 11 resulted in a 2.5-fold increase of Dfd–Exd affinity, it produced the opposite effect in Scr–Exd complexes by displaying a 4.5-fold decrease in affinity. Thus, these results indicate that the Hox–Exd binding sites present in *AP-2* enhancers are fine-tuned to provide maxillary-specific activation of *AP-2* by Dfd as well as to provide *AP-2* domain-specific activation by Dfd within this segment. Additionally, the non-conserved region separating domains 2 and 3 is required for the activity of the *AP2x-377* maxillary enhancer, as domain 3 by itself does not activate gene expression (Fig. 2b', h'). This could be due to additional cofactors interacting with the non-conserved region or sequence-driven conformational cues facilitating cooperative interactions between proteins in domain 3.

The identification of Dfd–Exd high affinity-binding sites present in *AP2x-377* was unexpected. However, converting these sites to high-affinity class 2 sites resulted in a 2–2.5-fold increase in Dfd–Exd affinity. These differences in Dfd–Exd binding affinity were sufficient to induce ectopic activation of the *AP2x-377* element in anterior and ventral regions of the maxillary segment, overlapping the primordia of Dfd-regulated maxillary structures, such as the dorsal cirri and mouth hooks. Moreover, *AP-2*[15] rescue experiments showed that expression of

*AP-2* in these primordia impaired the normal development of these structures. Therefore, these results suggest that Dfd–Exd complexes rely on small variations in binding affinity to control the domain-specific activation of their target genes. This ultimately orchestrates the deployment of the diverse morphogenetic programs establishing the identity of the maxillary segment (Fig. 9). The ectopic activation of optimal *AP-2* enhancers results from a higher sensitivity to lower concentration levels of Dfd in the anterior and ventral regions of the maxillary segment. Furthermore, activating Dfd expression throughout the embryonic ectoderm resulted in *AP2x-377-D23H* enhancer activity outside the maxillary segment, which was not the case for *AP2x-377*, showing that the *AP2x-377-D23H* enhancer is more sensitive to Dfd levels than *AP2x-377* (Supplementary Fig. 10). Similar results were obtained by modulating the levels of Dfd in the maxillary segment. Increasing the levels of Dfd in the anterior and ventral part of the maxillary segment resulted in the ectopic activation of *AP2x-377* in a similar pattern as the *AP2x-377-D3H* enhancer containing class 2 Dfd–Exd binding sites. In contrast, reducing Dfd levels in the anterior part of the maxillary segment reduced activation of *AP2x-377-D23H* to a pattern similar to the wild-type *AP2x-377* enhancer. Altogether, these results showed that the Dfd–Exd binding sites in *AP2x-377* are under tight constraints despite being of high affinity, as small variations in affinity can impair the development of several Dfd-regulated maxillary structures. Moreover, Dfd–Exd binding sites may have evolved to respond to the different concentration levels of Dfd throughout the maxillary segment. In dorsal cirri and mouth hooks cells, the Dfd concentration, while sufficient to trigger the genetic programs responsible for the development of these maxillary structures, is insufficient to activate the expression of *AP-2* (Fig. 9). Thus, we speculate that Dfd–Exd sites evolved to maintain the *AP-2* maxillary enhancer inactive in regions of low Dfd concentrations. To ensure activation of *AP-2* in the posterior border cells of the maxillary segment, the transcription regulation of Dfd must have co-evolved to ensure a higher concentration of Dfd in these cells. Indeed, expression of Dfd in the maxillary segment is regulated by a specific CRM that exhibits higher activity in *AP-2* expressing cells[29]. Moreover, this enhancer contains class 2 high-affinity core sequences required for its activation in the maxillary segment[30]. Thus, the variability of Dfd–Exd sites in the *AP-2* maxillary enhancer may have developed as part of a mechanism that allows for Dfd to cell-specific activate its target genes and coordinate the morphogenesis of the maxillary segment.

The general view in transcription is that low-affinity TF binding sites are regulatory points required for tissue/cell-specificity, while high-affinity sites are likely to be important if specificity is not required[31]. In the case of Hox TFs, the reported studies are in agreement with this view, with low-affinity Hox-Exd sites being required for Hox specificity[12]. However, our study of the Dfd-regulated maxillary *AP-2* enhancer, *AP2x-377*, unveils a new perspective of the role of Hox–Exd high-affinity binding sites in Hox specificity. Our results show that Dfd–Exd complexes, contrary to our initial predictions, bind with high affinity to a wider range of binding sites. Nonetheless, these sites direct the domain-specific activation of the *AP2x-377* element in the

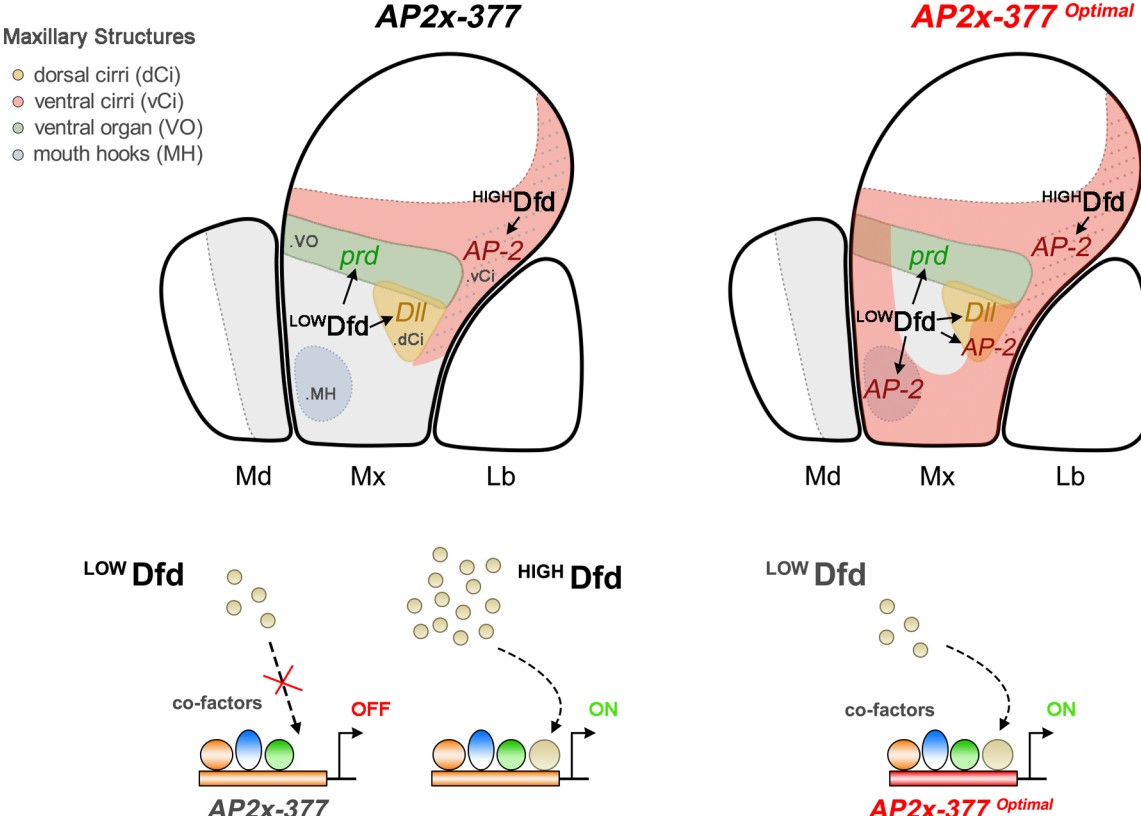

**Fig. 9 | The domain-specific activation of Dfd target genes in the maxillary segment relies on the balance between its concentration and binding affinity.** In wild-type embryos, *AP-2* is activated in the posterior maxillary border cells with high Dfd levels (HIGHDfd), enabling Dfd–Exd complexes to bind and activate the *AP-2* via the maxillary *AP2x-377* enhancer. In the remaining regions of the maxillary segment, Dfd is expressed at lower concentrations (LOWDfd) and is unable to activate *AP2x-377*. Optimization of Dfd–Exd sites in *AP2x-377* (*AP2x-377optimal*) results in an increase in Dfd–Exd binding affinity, which allows activation of *AP2x-377optimal* in the anterior and ventral regions of the maxillary segment. Co-factors of Dfd involved in the activation of the *AP2x-377* are present throughout the maxillary segment with the concentration and binding affinity of Dfd–Exd determining the domain-specific activation of *AP2x-377*. Primordia of Dfd-regulated maxillary structures: ventral organ (green area); dorsal cirri (yellow area); ventral cirri (pink area); mouth hooks (blue area). The grey area in the mandibular and maxillary segments (outlined by dash lines) indicates the area of Dfd expression. The dotted lines in the maxillary segment indicate the posterior border cells with high Dfd levels. Md Mandibular segment, Mx Maxillary segment, Lb Labial segment.

maxillary segment only and not in any other segments, showing that these sites are under tight constraints to achieve this dual specificity: precise activation in specific maxillary cells and no additional activation in other segments. Thus, Hox specificity does not seem to be a simple issue of low versus high-affinity Hox–Exd binding sites. Instead, these sites possess the appropriate affinity to allow one Hox TF to coordinate multiple events in one segment while remaining unresponsive to other Hox TFs. The fate map of the maxillary segment has been well characterized and Dfd targets involved in the development of the different maxillary structures have been determined (Supplementary Figs. 2)[28,32–34]. Although maxillary enhancers have been identified for some of these targets, such *paired* and *Dll*, required for the formation of the ventral organ and the dorsal cirri, respectively[32,34], these regulatory regions have not been studied in detail. As the balance between binding site affinity and Dfd concentration seems to play an important role in Hox specificity in the maxillary segment, further study of these different CRMs and the characterization of its Dfd-binding sites will shed more light on the generality of Hox–Exd high-affinity sites in cell-specificity and morphogenesis.

Our study of the Dfd-regulated *AP-2* enhancer challenges the current view that Hox specificity is achieved with low-affinity binding sites. This is consistent with a very recent study showing that the anterior Hox protein Sex combs reduced (Scr) in combination with its cofactor Exd recognize Scr–Exd high-affinity sites to specifically control target genes in the leg primordium of thoracic segment 1[35]. Thus, it seems that Hox TFs employ a wider range of mechanisms to ensure the specific

activation of their targets. Such mechanisms may have arisen due to the ability of Hox TFs to recognize a variety of binding sequences. The Hox binding sites bound by posterior Hox TFs are constrained by the promiscuous nature of the anterior Hox proteins[12]. They use low-affinity sites to achieve specificity in controlling posterior target genes. However, anterior Hox proteins do not only bind more sequences but these sequences also present high specificity toward anterior Hox, even when converted into so-called high-affinity binding sites. How can this be explained? It is possible that during evolution the appearance of new Hox TFs may have required the development of innovative molecular mechanisms to ensure a specific function of these TFs. In the case of the anterior Hox TFs, during the evolution of the head (anterior) Hox proteins may have acquired new regulatory features to allow the development of head-specific structures and functions. Interestingly, it has been hypothesized that during cephalization several of the anterior-most trunk segments shifted anteriorly, and some of these cephalized trunk segments (including the gnathal segments) adapted to feeding in the course of evolution[36]. Such a modification of a generic segmental program in these early organisms might have required anterior Hox proteins like Dfd to recognize a wider range and more specific binding sequences to be able to control head-specific morphologies and functions while at the same time keeping their ability to control "ground-state" genes active in all segments.

In future, it will be interesting to study whether similar principles identified in this study also apply to other homeodomain TFs, which constitute one of the largest TF classes in the animal kingdom.

## Methods

### Fly line and materials

With exception of the Dfd RNAi and Dfd overexpression experiments (Fig. 7 and Supplementary Fig. 10, respectively) that were performed at 29 °C, all experiments were done at 25 °C. The following transgenic lines and mutant alleles were used: *UAS-LacZ* (from A.Tugores), *UAS-Dfd* (from W. McGinnis), *Dfd[w21]* (from W. McGinnis), *69B-Gal4* (Bloomington *Drosophila* stock center #1774), *Abd-B[M1 37]*, *UAS-AP-2.PB* (Bloomington *Drosophila* stock center #23884), *AP-2[l5]* (Bloomington *Drosophila* stock center #23721), *UAS-myrRFP* (Bloomington *Drosophila* stock center #7118), *ptc-GAL4* (Bloomington *Drosophila* stock center #2017). The *AP2x-mCherry* reporter line was previously described[14]. For the *AP-2[l5]* rescue experiments, an *AP-2[l5], Abd-B[M1]* recombinant was generated to easily identify and quantify *AP-2[l5]* homozygous larvae. All transgenic lines were generated via PhiC31-site-directed transgenesis by Bestgene Inc. The transgenes were inserted on the 2nd chromosome by injecting $y^1 w;^{67c23} P\{CaryP\}attP40$ embryos.

### Plasmid constructs

*AP2x* enhancers were generated by PCR reaction (single-step or via a two-step overlapping reaction) as described in Supplementary Tables 3 and 4. Gateway entry plasmids were generated by subcloning the *AP2x* enhancers in the *pENTR/D-TOPO* vector using the *pENTR/D-TOPO* Cloning kit (Invitrogen). The resulting plasmids were recombined with the appropriate Gateway destination vector using the LR Clonase Enzyme Mix (Invitrogen). The following Gateway destination vectors were used: *Vanglow-GL-Luc:GFP* (addgene, plasmid #83338; for reporter lines directing expression of the recombinant protein Luciferase:GFP), *pBPGUw* (addgene, plasmid #17575; for reporter lines directing expression of Gal4), *pBPGUw-mCherryStrep* (for reporter lines directing expression of mCherry)[15] and *pBPGUw-mCD8-GFP* (for reporter lines directing expression of mCD8-GFP). The *pBPGUw-mCD8-GFP* plasmid was generated by replacing the Gal4 CDS in *pBPGUw* with the *mCD8-GFP* CDS.

pET24-His-Flag-Hth and pET24-His-myc-Exd were cloned in pET24 as followed: pUAST-myc-Exd and pUAST-Flag-Hth (a gift from James C-G Hombria) were used as templates to amplify by PCR DNA fragments containing the His-myc-Exd and His-Flag-Hth ORFs (see Supplementary Table 4 for primers). These fragments were digested with KpnI and subcloned in pET24. The pET24 plasmid was prepared from pET24-GFP. To remove the GFP ORF and subclone the His-myc-Exd and His-Flag-Hth ORFs, pET24-GFP was digested with NcoI, filled-in with Klenow to generate blunt extremities, and then digested with KpnI.

GST-Dfd was subcloned in the pGEX-6P[38]. To generate GST-Scr, the EST LD21370 in the pOT2 vector was used as a template and subcloned in the pGEX-6P[36].

### Immunohistochemistry and RNA in situ hybridization

The following primary antibodies were used: guinea pig anti-Dfd (1:500)[38], mouse anti-En 4D9 (1:2.5) (Developmental Studies Hybridoma Bank), rabbit anti-Paired (1:1000, from Markus Noll), rabbit anti-Dll (1:100, from Sean Carroll), mouse anti-βGal (1:1000, Ref: Z3781, Lot: 0000393241, Promega), rabbit anti-GFP (1:300, Ref: A11122, Lot: 2083201, Invitrogen) and rat anti-RFP (5F8) (1:100, ref: 5f8-100, Lot: 90228062AB-15, Chromotek).

For confocal microscopy, secondary antibodies were conjugated to Alexa Fluor 488 (mouse: Ref: 715-545-150, rabbit: Ref: 711-545-152, rat: Ref: 112-545, guinea pig: Ref: 706-545-148), Cy3 (mouse: Ref: 115-165, rabbit: Ref: 111-165, rat: Ref: 112-165, guinea pig: Ref: 706-165-148) and Alexa Fluor 647 (mouse: Ref: 715-605-151, rat: Ref: 712-605-150, guinea pig: Ref: 706-605-148) (Jackson ImmunoResearch Laboratories, 1:200). For widefield microscopy an anti-rat biotinylated antibody (1:100, Ref: 112-065-003, Jackson ImmunoResearch Laboratories) was used as a secondary antibody.

For fixation, embryos were collected and dechorionated in 50% (v/v) bleach for 5 min, rinsed in water, and transferred to a 1:1 heptane/formaldehyde 6% (v/v) (in PBS) solution for 30 min. The formaldehyde was then removed and replaced by an equal volume of methanol. The vitellin membrane was removed by agitating the embryos for 60 s. The devitenillinized embryos were collected and washed two times with methanol.

For immunostaining, embryos were rehydrated in PBT (PBS + 0.1% (v/v) Tween 20) for 30 min and then blocked in PBT + 1% (w/v) BSA for another 30 min. The embryos were then incubated with primary antibodies for 1 h at room temperature or overnight at 4 °C. The embryos were then washed 4 × 15 min in PBT + 1% (w/v) BSA, incubated with secondary antibodies in PBT + 1% (w/v) BSA for 1 h at room temperature, and again washed 4 × 15 min in PBT. For confocal microscopy analysis, the embryos were mounted in VectaShield and analysed with a Leica TCS-SP8 confocal microscope. For Widefield analysis, the Peroxidase reaction was developed using the VECTASTAIN Elite ABC-HRP Kit (PK-6100, Vector Laboratories) and the DAB Peroxidase Substrate Kit (SK-4100, Vector Laboratories). The embryos were then dehydrated for 5 min in 30% (v/v) Ethanol, 5 min in 50% (v/v) Ethanol, 5 min in 70% (v/v) Ethanol and in 100% (v/v) Ethanol. The embryos were then mounted in Durcupan and analysed using a Zeiss Axio Imager M1 microscope.

For in situ Hybridization, the embryos were rehydrated 2 × 15 min in PBT, followed by a 30 min fixation in formaldehyde 6% (v/v) (in PBS). The embryos were then washed 3 × 2 min in PBT and incubated for 1 min in Proteinase K at 20 ng/µl. The reaction was stopped by washing 2 × 2 min with Glycine at 2 mg/ml in PBT followed by 2 × 5 min in PBT. The embryos were then fixed again in formaldehyde 6% (v/v) in PBS and washed 5 × 2 min in PBT, 10 min in 100 µl of Hybridization solution/ PBT (1:1) and 10 min in 100 µl of Hybridization solution. Afterwards, the embryos were incubated in Hybridization solution for 1 h at 55 °C after which the RNA probe (denatured at 95 °C for 5 min) was added with incubation proceeding overnight at 55 °C. The next day the embryos were washed at 55 °C as follows: 2 × 20 min in the Hybridization solution, 20 min in Hybridization solution/PBT (8:2), 20 min in Hybridization solution/PBT (6:4), 20 min in Hybridization solution/PBT (4:6) and 20 min PBT at 55 °C. The embryos were then washed at room temperature in PBT for 20 min and incubated for 1 h with anti-DIG alkaline phosphatase conjugated antibody in PBT (1:2000, Roche). Afterward, the embryos were washed 4 × 15 min in PBT followed by 3 × 15 min washes in alkaline phosphatase buffer (100 mM NaCl, 50 mM MgCl₂, 100 mM Tris–HCl pH 9.5). The embryos were then transferred to 1 ml of alkaline phosphatase buffer and the in situ was developed by adding 20 µl of NBT/BCIP (Roche). The reaction was stopped by rinsing the embryos with ice-cold PBT. The embryos were then dehydrated for 5 min in 30% (v/v) Ethanol, 5 min in 50% (v/v) Ethanol, 5 min in 70% (v/v) Ethanol and finally in 100% (v/v) Ethanol. The embryos were then mounted in Durcupan and analysed using a Zeiss Axio Imager M1 microscope.

Hybridisation solution: 5 × SSC, 50% (v/v) Formamide, 100 µg/ml salmon sperm DNA, 40 µg/ml *E.coli* tRNA, 50 µg/ml heparin and 0.1% (v/v) Tween 20.

All microscope images (confocal and widefield) were processed using Adobe Photoshop CS6 software. Fluorescence quantification was performed with Fiji.

### Quantification of fluorescence levels

To determine the activity of the different *AP2x-377*-derived enhancers, embryos were imaged with a ×63 objective in a Leica TCS-SP8 Confocal microscope and fluorescence levels were quantified with Fiji using the ROI manager tool. The activity was determined by measuring the β-galactosidase/Dfd fluorescence ratio. In Figs. 5 and 6, the β-galactosidase/Dfd fluorescence ratio was measured in the posterior-ventral border cells of the embryonic maxillary segment (between 20 and 39

cells per maxillary segment). The overall average for each embryo was determined and the results were plotted with Graphpad Prism 9 using a one-way ANOVA statistical test to determine the corresponding mean and standard error (Figs. 5, 6) or processed in Excel and tested by a two-sided $t$-test (Fig. 7). The box plots were generated in R.

## Cuticle preparation and cirri analysis

Embryos with 0–2 h of development at 25 °C were collected and developed for an additional 18 h at 25 °C. The embryos were dechorionated in 50% (v/v) bleach for 5 min, rinsed in water and transferred to 1:1 heptane/methanol solution, and agitated vigorously for 60 s. The larvae were then collected and washed two times with methanol and two times with water containing 0.1% tween. The larvae were mounted in Hoyer's medium and incubated for 3 days at 60 °C. Cuticles were analysed with a Zeiss Axio Imager M1 microscope and/or a Leica TCS-SP8 confocal microscope. Images were processed using Adobe Photoshop CS6 and Leica Application Suite X (LAS X) 3D Visualization software. The quantification of the dorsal cirri was plotted using Graphpad Prism 9 and an ordinary one-way ANOVA statistical test was used to determine its statistical significance.

## Protein expression and purification

The recombinant proteins were produced from BL-21 (RIPL) bacterial strain and purified on Ni-NTA agarose beads (Qiagen) or Glutathione-Sepharose beads (GE-Healthcare). To quantify the purified proteins, a standard curve was made by running in a 10% SDS–PAGE known concentrations of bovine serum albumin (BSA) together with dilutions of the produced proteins. The gel was then stained with Coomassie and the proteins were quantified using Fiji (Analyse/Gel). For the standard, a trend line equation was calculated in Microsoft Excel and this equation was used for calculating the protein amounts of the respective elution fractions.

## Electrophoretic mobility shift assays

Single-stranded complementary 5'-Cy5-labelled oligos were synthesized (Eurofins) and annealed in Annealing Buffer (10 mM Tris–HCl pH 8.0, 1 mM EDTA pH 8.0, 50 mM NaCl) to a final concentration of 1 pmol/µl (see Supplementary Table 5 for primer sequences).

Binding reactions were performed on ice for 20 min in a volume of 30 µl containing 1x binding buffer (20 mM HEPES pH 7.9, 1.4 mM $MgCl_2$, 1 mM $ZnSO_4$, 40 mM KCl, 0.1 mM EGTA, 5% Glycerol), 200 ng Poly(dI-dC), 3 µg BSA, 10 mM DTT, 0.1% (v/v) NP40, 2 pmol of double-stranded 5'-Cy5-labelled oligo and the selected proteins. 1% (w/v) agarose-TBE gels were prepared and pre-run in 1xTBE at 4 °C for a minimum of 30 min at 90 V. The binding reactions were then loaded on the gels and run at 4 °C, for 2 h at 90 V. Cy5-labelled DNA–protein complexes were detected by fluorescence using an INTAS Imager. For EMSA quantifications, each experiment was performed twice and raw image files were measured using Fiji (Analyse/Gel). The shifted band of complexes as well as the band of unbound free probes were used for the calculations of the bound fraction (% bound = shifted_band/(shifted_band + free_probe)*100). The mean and corresponding standard deviation for these values was plotted using Graphpad Prism 9 and a two-way ANOVA statistical test was used to determine its statistical significance. For $K_D$ calculations, the software Graphpad Prism 9 was used (non-linear regression/binding saturation/one_site-specific_binding).

## ChIP experiments

ChIP experiments were performed as described in ref. 39 and analysed by using qPCRs (Invitrogen Syber-Green-Mix). The following antibodies were used: gp-Dfd[38] (2 µl), gp-IgG (2 µl normal IgG, Ref: A82266, antibodies.com). ChIP protocol in brief: Embryos were collected, the chorion removed and cross-linked in cross-linking solution (50 mM HEPES pH 8, 1 mM EDTA, 0.5 mM EGTA, 100 mM

NaCl, 1.6% formaldehyde) or 15 min shaking. The reaction was stopped by transferring the embryos in a 15 ml tube, adding 10 ml PBT/glycine (500 µl 2.5 M glycine in 10 ml PBT), and vigorously shaking for at least one minute. The embryos were collected by centrifugation (1000×$g$ for 2 min) and washed twice with cold PBT, afterwards, they were frozen in liquid nitrogen and at −80 °C. The chromatin was prepared as the following: about 700–800 g of embryos were thawed and resuspended in cold PBT. The embryos in PBT were crushed to maintain cells. The cells were collected by two centrifugation steps, 400×$g$ for 1 min at 4 °C to remove cell parts and 1100×$g$ for 10 min at 4 °C to pellet the cells. The cells were lysed in cell lysis buffer (5 mM HEPES pH 8, 85 mM KCl, 0.5% NP40, protease inhibitors) and crushed. The resulting nuclei were collected by 2000×$g$ for 4 min at 4 °C. The nuclei were lysed in nuclear lysis buffer (50 mM HEPES pH 8, 10 mM EDTA, 0.5% N-Laurylsarkosin, protease inhibitors) for 20 min at room temperature. The chromatin was sheared in a Bioruptor (30 s ON/30 s OFF, 18×, high mode), afterward centrifuged at 20,000×$g$ for 10 min at 4 °C, aliquoted in 200 µl aliquots and frozen in liquid nitrogen and at −80 °C. The IP was performed by using an aliquot of the chromatin for the specific pull down and one for the MOCK (IgG). The pull-down used a mix of Sepharose beads A and G (Invitrogen). The beads were washed in RIPA Buffer (140 mM NaCl, 10 mM Tris–HCl pH 8, 1 mM EDTA, 1% TritoX100, 0.1% SDS, 0.1% sodium deoxycholate (DOC, make 10% DOC stock immediately before use), 1 mM PMSF, protease inhibitors). The chromatin was thawed, moved to Lobind tubes (Eppendorf) and filled to 500 µl with dilution buffer (4% glycerol, 10 mM Tris–HCl pH 8, 1 mM EDTA, 0.5 mM EGTA, protease inhibitors). The following ingredients were added to the chromatin solution: 100 µl 10% TritonX100, 100 µl 1% DOC, 100 µl 1% SDS, 100 µl 1.4 M NaCl, 10 µl 100 mM PMSF to mimic the RIPA buffer conditions. The prepared chromatin was precleared for 1 h with prepared Sepharose beads. The beads were collected by 1000×$g$ for 2 min at 4 °C. The supernatant was removed in a new tube, 1% input was separated and the rest incubated rocking with 1–3 µl antibody overnight at 4 °C. In addition, a new set of Sepharose beads were prepared and incubated rocking in RIPA + 1 mg/ml BSA overnight at 4 °C. The next day the antibody-chromatin solution was incubated for 3 h with the prepared BSA-beads and washed for 10 min in the following solutions, each time the beads were collected by 1000×$g$ for 2 min at 4 °C. Washing steps without extra proteins inhibitors: once in RIPA Buffer, 4 times RIPA buffer containing 0.5 M NaCl (500 mM NaCl, 10 mM Tris–HCl pH 8, 1 mM EDTA, 1% TritonX100, 0.1% SDS, 0.1% DOC, 1 mM PMSF), once with LiCl (250 mM LiCl, 10 mM Tris–HCl pH 8, 1 mM EDTA, 0.5% NP40, 0.5% DOC), two times with TE (10 mM Tris–HCl pH 8, 1 mM EDTA). After the IPs and the input were resuspended in 100 µl TE and 50 µg/ml RNase A was added to all samples and incubate for 30 min at 37 °C. Afterward, each sample was adjusted to 0.5% SDS and 0.5 mg/ml proteinase K and incubated at 37 °C ON. The next day the chromatin was de-cross-linked at 65 °C for 6 h and the DNA was purified by using Phenol/Chloroform (Invitrogen) and 5Prime gel tubes.

ChIP-qPCR results were calculated as described in ref. 40. Primers were located in the transgene priming the inserted enhancer (AP2x-377-int_For2: ACGCGCGTTCAACATTTAGG) and the vector (AP2x-377-int_Rev1: TTTATACCGCTGCGCTCGAT). Control primers were located in a region on the 2nd chromosome with no Hox binding sites (2L-FW: aggtgttgttgtgtgggtcctt, 2L-RW: tcccagagttcccttttagca). Calculation in brief: The IP and mock samples were normalized to the Input, taking into account that the input has 1% (removing 6644 cycles). The results were used to calculate the ΔΔCt value and then the resulting $2^{-ΔΔCt}$. The value of the specific primer (transgene) was normalized against the control (2 L) to obtain the fold enrichment. The calculation was performed in Excel and illustrated in R. Statistical significance was tested with the $t$-test.

## Computational analysis

The analysis of Hox-Exd binding sites in *AP2x-377* was performed using the No Reads Left Behind (NRLB) algorithm (https://github.com/BussemakerLab/NRLB)[24]. To perform the conservation analysis of *AP2x-377*, the DNA sequences from the different *Drosophila* species were obtained from the UCSC Genome browser (http://genome.ucsc.edu) and used to perform a Mauve Alignment with the Geneious 9.1.8 software. The alignment of the Dfd-Exd binding sites present in *AP2x-377* was done using the alignment tools in Geneious 9.1.8 software.

## Reporting summary

Further information on research design is available in the Nature Research Reporting Summary linked to this article.

## Data availability

Source data

The source data underlying Figs. 4, 5, 6, 7, 8 and Supplementary Figs. 6, 7, 8, 11 are provided as Source Data file. Source data file is underlined in the legend of all referred figures. Other raw files are available upon reasonable request. Source data are provided in this paper.

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

## Acknowledgements

We thank the Bloomington Center for fly lines, the DHSB for antibodies and Addgene for plasmids. We thank you for sharing their materials, James C.-G. Hombria (Exd and Hth plasmids), Sean Carroll (antibody anti-Dll) and Markus Noll (antibody anti-Prd) as well as the Bloomington Drosophila Stock Center for fly lines, the Developmental Studies Hybridoma Bank for antibodies and Addgene for plasmids. We further thank Patrick Van Nierop y Sanchez for critically reading the manuscript. This project was supported by the Deutsche Forschungsgemeinschaft (DFG LO 844/5-2). For the publication fee we acknowledge financial support by Deutsche Forschungsgemeinschaft within the funding programme „Open Access Publikationskosten" as well as by Heidelberg University.

## Author contributions

Conceptualization: P.B.P. and I.L. Experimental design: P.B.P. and I.L. Experimental procedures: P.B.P., M.W., J.C., X.G. Data analysis: P.B.P., K.D., X.G. Computational analysis: P.B.P. and K.D. Writing: P.B.P. and I.L. Funding acquisition: I.L.

## Funding

## Competing interests

The authors declare no competing interests.
