## [Peer Review File · Nature Communications]

Specificity of the Hox member Deformed is determined by transcription factor levels and binding site affinitiesREVIEWER COMMENTS

Reviewer #1 (Remarks to the Author):

This manuscript by Pinto, P.B., et al. reports data pertaining to the so-called Hox paradox, which consists in the paradoxical observation that Hox transcription factors fulfill specific functions in vivo while sharing highly similar homeodomains and DNA binding in vitro. While this paradox might partly be resolved by interacting proteins (like TALE homeodomain proteins) which modulate the DNA binding specificity or elicit latent specificity for the DNA binding of distinct Hox proteins, the importance of DNA binding affinity for distinct sequences has also been at the heart of this problematic. Hox proteins expressed in posterior domains of the developing drosophila embryo have been shown to deploy functional specificity by the recognition of multiple low affinity (or sub-optimal) DNA sequences. In this manuscript, the authors address if this observation can be generalized to the so-called "anterior" Hox proteins, ie Hox proteins active in anterior segments. The conclusions the authors reach is that anterior and posterior Hox transcription factors have developed different mechanisms to ensure their functional specificity, in particular that for anterior Hox proteins, not for posterior ones, high affinity sites can contribute to highly specific target gene regulation. Although the important set of data presented here, combining biochemical evidence (EMSAs), spatial target gene regulation in the embryo (reporters) and genetic analyses (phenotypes), the study is focused on only one Hox protein, Dfd, and one target enhancer, with multiple DNA target sites, AP2x. Thus, although the results are clear, it is maybe too preliminary to generalize them to other anterior Hox proteins and other target genes. It cannot be excluded actually that in other contexts, other target genes are specifically regulated by anterior Hox proteins according the same "rule" as the one already described for posterior Hox proteins, ie on the ground of combined low-affinity sites. Therefore, the title of the submitted manuscript is somehow too general, since the data provided do not allow to conclude for a "cis regulatory code" (what does "code" mean in the context of the study here?), for "anterior Hox specificity". The title of the article must be changed to better fit with the conclusions of the work.

Similarly, in the body of the manuscript some conclusions should be qualified. For example, at the end of the introduction the authors conclude "In sum, our results demonstrate ...that anterior and posterior Hox TFs have developed different mechanisms to ensure such specificity" is an overstatement, it is an inappropriate generalization of what can be concluded from the data. Other example: in the discussion section, the authors write that their "study ...supports the observation that anterior and posterior Hox TFs rely on different mechanisms to achieve Hox specificity". This is also an excessive generalization of the results. It cannot be excluded that anterior proteins could achieve their specificity like posterior Hox in distinct contexts, on distinct targets. So, I would suggest the authors to avoid too broad conclusions. Nonetheless, it is true that the data challenge the view based on posterior Hox protein studies, that low affinity binding sites are the rule to ensure Hox protein specificity.

The authors carried out multiple EMSAs to test putative Dfd-Exd sites. Interestingly, they did not find a clear correlation between the predicted relative binding affinity and the observed binding efficiency (eg page 7 Oligo C vs Oligo A). Overall, all sites predicted to be of low affinity were actually bound with "comparable high affinities". With reference to what standard the authors can qualify the binding as with "high affinity"? That the correlation between predicted binding and observed binding is poor is one thing, qualifying the observed binding as of high affinity needs to be justified and objectified. This is critical since one key message of the work is that "high affinity sites can act highly specific" in the mode of action of anterior Hox proteins. In the bulk of the "EMSA" section of the manuscript the authors indicate, for example, "...Oligo D...harbours sites with a rather high relative affinity...", "...a similar relative affinity to other sites...", "...Dfd-Exd sites were all bound rather efficiently..." But at the end of this section the authors conclude "...taken together...all of which were bound ...with comparable high affinities." We can in fact compare the relative binding of Dfd-Exd onto sites by looking at EMSAs and their quantification (but see comment about quantification below), while qualifying these as "high affinity" binding requires reference controls and if possible K_a quantification.

Now, when the authors state that (end of section page 7) "...experiments identified multiple Dfd-Exd binding sites....all of which were bound by Dfd-Exd complexes with comparable high affinities", this is

also not completely correct since, according to the oligos designed, all sites have not been tested individually. Indeed (cf Fig.3 and suppl Fig. 7), sites 3, 5, 7 or 11 have been analyzed (mutated) separately, while not sites 9 and 8.1/2/3, sites 2 and 4 or sites 1 and 6. The fact that some predicted sites overlap makes drawing conclusions obviously difficult about individual sites. But this calls for a comment while describing/discussing EMSAs.

While "optimizing" core binding sites, the authors observed a modest increase in the binding of Dfd-Exd. This however has been observed while involving 120 ng of Dfd (Suppl Fig. 8 b). It is possible that a more differential binding would have been observed while adding lower amount of proteins. To conclude that optimizing the sequence has a modest effect on affinity requires testing a certain range of protein concentration to show that affinity/dissociation curves almost overlap. This is really important in particular since the authors highlight a discrepancy between the modest effect of optimizing the core (and flanking sequences) and the fact that the optimized sequences display a significantly different response (reporter, phenotype) according to low- and high-Dfd concentration territories. If the AP2x-377 and AP2x-377-D23H enhancers are sensitive to Dfd levels (Fig.6), and if the optimized enhancer supports spatial mis-regulation in low-Dfd cells (and impairs maxillary structures development), I would suggest the authors to repeat EMSAs on wt versus optimized sequences with distinct Dfd amounts to possibly resolve this discrepancy.

Related to this, how the proteins involved in EMSAs were quantified needs clarification. In the Methods section the authors indicate that proteins were quantified "using Fiji". Is this method reliably and reproducibly quantitative? On some EMSAs (eg Figure 3e or Suppl. Fig. 7b), the amount of Dfd involved is indicated (38 ng, 76 ng, ...). To what extent are these values reliable, have they been determined according to reference samples of known concentration? Quantification of proteins requires clarification. This is important to support the conclusion that Dfd binds sites with "comparable high affinity" and that "optimizing the sequence has a modest effect on affinity".

Comparing the expression patterns of the AP2x-377, -D3H and -D23H reporters, the authors indicate (page 9, cf Figure 5) that the -D3H and -D23H reporters display similar ectopic expression, supporting that Domain 3 of the AP2x enhancer is crucial. While looking at Figure 5 c', d' and f', it seems that the ectopic expression provided by D23H is more extended and, visually, the pattern provided by -D3H appears more similar to the wt AP2x-377 than to -D23H.

Along the same line, in the discussion section (page 13, §2) the authors state that "...the AP2x-377-D23H enhancer is more sensitive to Dfd levels than AP2x-377." This cannot be affirmed based on the data since the wt and -D23H enhancers might be sensitive to different ranges of Dfd concentrations, which has not been addressed.

Minor points

In the abstract, the "TF" abbreviation is not defined

Introduction, page 3, the authors indicate "...Hox TFs share the same DNA-binding domain, the homeodomain". I would indicate "highly similar" or "extremely conserved" rather than "the same". While identifying the sequence determinants of the AP-2 enhancer activity, the authors conclude that the "AP2x-377 represents the minimal enhancer for Dfd-dependent activation". This is not precisely correct, AP2x-377 represents the minimal enhancer tested providing Dfd-dependent activation, thus it contains the minimal information but does not define the minimal enhancer. More sequence dissection could lead to a more refined definition of the necessary regulatory elements defining the enhancer. So, I would suggest to rephrase this sentence "In sum, these results revealed that... represents the minimal enhancer...." on page 6.

Looking at Figure 2 presenting reporter expression patterns, Fig. 2 b and d seem to show that the AP2x and AP2x-377 enhancers do not provide reporter activity in identical territories. The AP2x-377 enhancer seems to drive ectopic expression at st11. This appeals for a comment in the text, pages 5-6. Similarly, the AP2x-230 and -268 enhancers still promote some signal (Fig. 2f, g) while the authors indicate that the deletions made to generate these -230 and -268 enhancers "resulted in loss of enhancer activity in earlier stages". Does the signal visible on Fig. 2f and g correspond to background ectopic signal?

While describing EMSAs, the authors should provide more details about quantification. Graphs appearing on Fig.3 c, f, g and Suppl. Fig.7 c, f, g present "Binding %", but what defines the 100%,

how the quantification is normalized should be indicated.

Next on Fig.3 b, e and Suppl Fig. 7b, e, EMSA gels are presented with two frames each, a wide, top panel showing non-retarded and retarded bands, bottom panels appearing as the non-retarded bands but with lower exposure, is that correct? A better description of what is presented should be provided (legend, maybe arrows or arrowheads to identify bands...).

In the discussion section, the authors indicate (page 12, §3): "Additional cofactors are required to interact specifically with the Dfd-Exd complexes bound to domain 3. This is supported by our observation that the non-conserved region...is required". Although it is indeed highly probable that additional factors may be required to interact with Dfd-Exd onto domain 3, this sentence is somehow too affirmative since non-conserved sequences in a regulatory sequence can be influential as such by facilitating the cooperative interaction between proteins, eg thanks to sequence-driven conformation of the DNA (DNA bending for example, opening of grooves, etc...).

Typos and clarifications:

*Page 7, §1, "Supplementary Fig. S7D-F" should appear as "Supplementary Fig. 7d-f"

*Page 9, §2, while referring to optimized reporters, the authors indicate "(Fig. 3a, d-d)", while it should be "(Fig. 5a, d-d)".

*In the Methods section, the authors refer to "mCherryStrep" reporter lines. This probably corresponds to the mCherry reporters referred to elsewhere in the manuscript. This needs to be clarified.

*Legend to Figure 1, a word seems missing in the "(c-e)" description: "AP2x activity of during embryogenesis", should maybe appear as "Activity of the AP2x enhancer during embryogenesis" ?

*Legend to Figure 1: what is described on pictures "d, d'" and "e, e'" should be clarified. The authors indicate "In stage 12, posterior ventral cells start to activate AP2x (red in d, grey in d') with full activation being achieved by stage 13 (red in d, grey in d') and maintained until the end of embryogenesis." Shouldn't this appear as "In stage 12, posterior ventral cells start to activate AP2x (red in d, grey in d') with full activation being achieved by stage 14 (red in e, grey in e')"?

*Legend to Figure 1, the authors refer to a "white dashed line in b, c and d" while such a line actually appears on "c, d and e"

*Legend to Figure 9, line 4 "LOWDFD" should appear as "LOWDfd"

*Legend to Supplementary Figure 2 needs to be revised. Descriptions pertaining to images are partly incoherent (confusion between "e" and "d": "e" does not represent expression pattern of the AP2x-mCD8-GFP reporter; "a" and "b", "a'" and "b'": Dll expression is observed on a-a" and Dll-2-5kb-mCherry on b-b"; what is indicated as "st17" embryos and "st16" embryos: "st16" corresponds to "c" and "st17 to "d", ...)

*Supplemental Figure 4 a-b", annotation referring to AP2x-377 should be changed into AP2x-377-mCherry. Supplemental Figure 4 c-d", annotation referring to AP2x-377 should be changed into AP2x-377-mCherry.

*Supplemental Figure 5 b, why are some characters in the alignment highlighted with yellow and brown backgrounds should be indicated in the legend.

*Legend to Supplementary Figure 10 needs to be revised. Descriptions pertaining to images are partly incoherent (confusion between c and d; images c', c'" and c'" do not exist on the figure; images e, e', e'" and e'" are not described in the legend...)

Reviewer #2 (Remarks to the Author):

The manuscript by Pinto et al. investigates the functional balance between transcription factor concentration and low affinity binding sites in development.

The authors identify a minimal AP2 enhancer, which drives appropriate AP2 expression in the maxillary region under control of the HOX transcription factor Dfd. The enhancers contain 'low affinity' Dfd binding sites, i.e. sites that do not match the expected sequence motif. Unexpectedly, these sites are strongly bound by Dfd in vitro and their 'optimization' to canonical sites only results in a slight

increase in Dfd binding affinity. Most intriguingly, despite the small increase in Dfd binding in vitro, a version of the enhancer with optimised sites displays ectopic expression and affect maxillary morphogenesis.

The authors conclude that the precise target gene regulation underlying Dfd-regulated morphogenesis relies on a 'balanced' co-evolution of Hox binding affinities and Hox protein concentration. The authors base their conclusion on high quality data, sound and beautifully presented. In many cases the conclusion is strengthened by independent approaches to fully evaluate the functional activity of HOX sites.

The key strength of this paper is in the integration of careful and detailed in vitro analyses with their corresponding functional analyses in vivo to assess how (small) changes in the biochemical activity of transcription factors translate into (dramatic) in vivo effects.

We do not understand yet how transcription factor binding in vitro translates on transcription factor activity in vivo. This is a crucial gap, which deeply undermines our ability to predict (and subsequently modify) how gene expression is controlled. This large disconnect is mainly due to the fact that very few studies take a holistic approach by simultaneously addressing these two aspects of transcription factor activity, binding in vitro and activity in vivo as presented in this paper. Addressing this disconnect is crucial to understand how transcription factors activity are converted in control of gene expression and eventually morphogenesis. Therefore, this study is highly significant for our understanding of how gene expression is controlled in vivo.

Main comments:

I have not found a clear and straightforward explanation of how the authors define high affinity sites. Are those based on in vitro studies? Clearly, their experiments show that low affinity sites are high affinity in vitro as well. What is the supporting evidence for high affinity sites? This is crucial to distinguish high and low affinity sites. Otherwise it risks becoming a semantic exercise...

The differences in Dfd in vitro binding after optimization of the sites are quite small as the authors have noted. Have they considered the possibility that mutagenesis of these sites rather than optimizing Dfd binding in vivo can decrease binding of a repressor (which is competing with Dfd for accessing the AP2 enhancer)?

Specific comments:

Several supplementary figures contain information that are essential to interpret the main figures. For example:

Figure 2: what are 1, 2 and 3 in blue and what are the yellow areas? I only understood that by looking at the corresponding supplementary figure

Figure 3: it was rather difficult (and time consuming) to piece together all the information required to fully interpret the results in this figure and their conclusion. Part of the information is in the result text and part in the supplementary figure. It was unclear how these sites were selected to be tested in EMSA, and where they are in the enhancer.

The authors should reorganise the figure by adding a drawing of the enhancer showing the location of the sites and merging with panels from supplementary figure 5 to show the sequence of the sites.

Figure 3:

The oligos used in EMSA should be renamed to reflect the nomenclature of the sites.

The authors should test a canonical Dfd motif to quantify the relative binding of Dfd to 'low affinity' sites

I assumed the probe was at the bottom of the main area outlined, but there is a smaller section outlined at the bottom of each gel - what is the band contained here? And what are the numbers underneath? I did not find any reference to those numbers in the text - if so they should be removed.

There is no description of how binding quantification were made.

Supplemental Figure 5: The graph shows different affinities – how does the search tool classify those identified as low affinity binding sites? If it defines sites as low affinity based on their distance from a high affinity site, does the search tool definition of high affinity site coincide with the one of the authors?

What does class 2DE in b means? And what are the yellow and brown highlights?

Figure 4: Can the expected phenotype be described for the non-specialist reader? Is it the number/shape of cirri that is rescued and/or is defective?

Figure 5: a. what does H mean?

Why is the strategy to assess enhancer activity different here? Instead of being upstream of a reporter (as in previous figures) the enhancer is placed upstream of GAL4 and the readout is obtained by crossing with UAS-lacZ strain.

Figure 6: The observation that optimization of Dfd sites increases the response of the enhancer to Dfd, but does not change enhancer specificity is very interesting. The authors should back up this information showing how Scr binds in vitro to enhancer exemplar low affinity sites and their optimised counterparts.

Figure 7/8: I suggest merging these two figures (possibly with the help of a supplementary figure) and move them immediately after Fig. 5 (Fig 6 interrupts the logic of the flow).

P4: This sentence is unclear:

“This simplistic view of binding site affinity and Hox specificity certainly holds true for posterior Hox proteins, as the activation of their target genes needs to take into account the promiscuous binding of the anterior Hox-Exd complexes that imposes constrictions to the posterior Hox proteins”.

Reviewer #3 (Remarks to the Author):

The authors examine Dfd binding to the AP2x enhancer in vitro and correlate this with enhancer function in vivo. Based on this information they propose a general model for the function of anterior Hox genes. I have several concerns that preclude me from supporting publication at this time.

1. The fact that many enhancers contain suboptimal binding sites and that optimization of such enhancers leads to abnormal function is well established.
2. I do not believe that one can use data from a single enhancer and a single Hox TF to generalize 'a cis regulator code for anterior Hox specificity' as indicated by the title. To do so, it would be necessary to first look more broadly at high/low binding sites for other anterior Hox genes.
3. The definition of 'low' and 'high' affinity sites is unclear. The use of the term 'low affinity' sites sometimes seems based on predictions in the literature and sometimes on their own EMSA data, which causes confusion. In fact, since the binding sites predicted to be low affinity (based on literature and NRLB) actually bind Dfd with high affinity, doesn't this simply suggest that the algorithm is faulty? To continue to refer to such sites as 'low affinity' seems disingenuous. It is also unclear how the authors define high versus low affinity in their EMSA assay. There seems to be no absolute control included, instead this seems to be defined in relative terms. For instance, on what basis is Dfd binding to the AP2x elements defined as 'high affinity'. Is this a post-hoc conclusion based on the fact that Dfd binds the suboptimal and optimal sites with the same affinity?
4. I am not convinced that the observed 1.3 fold difference in affinity is the causative variable in vivo. Considering that the chromatin state in vivo is dramatically different than in the EMSA, alternative possibilities can be readily envisioned. In fact, the lack of in vivo binding data (by CHIP etc.) is a weakness of the study.

5. The naming of the various domains, binding sites, mutants and oligos is very confusing. It is frequently very difficult to understand how a particular mutation relates to a specific binding site and the corresponding element of the AP2x enhancer.

POINT-BY-POINT RESPONSE TO THE REVIEWERS:

We would like to thank all the reviewers for their constructive criticism and helpful suggestions, which have helped to improve the manuscript significantly. We have included many new data, have performed new analyses and have substantially re-written the manuscript, which we describe in detail below.

Individual comments to the reviewers' questions:

Reviewer #1: thought the study provides an important data set with regards to Hox specificity, combining biochemical evidence (EMSA), spatial target gene regulation in the embryo (reporters) and genetic analyses (phenotypes). However, this reviewer also thought that the manuscript would benefit from some corrections before publication in Nat Comm. We addressed the comments of this reviewer in the following way:

MAJOR POINTS:

#1. ... although the results are clear, it is maybe too preliminary to generalize them to other anterior Hox proteins and other target genes. It cannot be excluded actually that in other contexts, other target genes are specifically regulated by anterior Hox proteins according the same "rule" as the one already described for posterior Hox proteins, ie on the ground of combined low-affinity sites. Therefore, the title of the submitted manuscript is somehow too general, since the data provided do not allow to conclude for a "cis regulatory code" (what does "code" mean in the context of the study here?), for "anterior Hox specificity". The title of the article must be changed to better fit with the conclusions of the work.

We agree with the reviewer's comment regarding the general view portrayed in the title and the lack of data supporting such statement. As a result, the title has been changed to "**A fine-tuned balance of transcription factor levels and binding site affinities determines specificity of the anterior Hox protein Deformed**" which we believe conveys more accurately the conclusions of our work.

#2. Similarly, in the body of the manuscript some conclusions should be qualified.

For example, at the end of the introduction the authors conclude "In sum, our results demonstrate ...that anterior and posterior Hox TFs have developed different mechanisms to ensure such specificity" is an overstatement, it is an inappropriate generalization of what can be concluded from the data.

We agree with the reviewer's comment that although our work challenges the view based on posterior Hox protein studies, that low affinity sites are the rule to ensure Hox protein specificity, there might be anterior Hox TFs that follow similar mechanisms as the posterior Hox TFs either in different contexts or on distinct target genes. Therefore, we changed the end of the introduction to clarify and summarize the conclusions of our work. Instead of "*In sum, our results demonstrate that Hox-Exd high-affinity sites can act highly specific in Hox target gene regulation and that anterior and posterior Hox TFs have developed different mechanisms to ensure such specificity.*", it now reads

"In sum, our results demonstrate that Dfd-Exd high-affinity sites can act highly specific in Hox target gene regulation, which challenges the view based on posterior Hox protein studies that low-affinity Hox/Exd sites are the rule to ensure Hox protein specificity."

Other example: in the discussion section, the authors write that their "study ...supports the observation that anterior and posterior Hox TFs rely on different mechanisms to achieve Hox specificity". This is also an excessive generalization of the results. It cannot be excluded that anterior proteins could achieve their specificity like posterior Hox in distinct contexts, on distinct targets.

We have changed this sentence so as to remove any over interpretation of our findings. To convey more accurately the conclusions of our work where it read “*Our study of the Dfd-regulated AP-2 enhancer supports the observation that anterior and posterior Hox TFs rely on different molecular mechanisms to achieve Hox specificity.*”, it now reads

“Our study of the Dfd-regulated AP-2 enhancer challenges the current view that Hox specificity is solely achieved by low-affinity binding sites. It seems that Hox TFs employ a wider range of mechanisms to ensure the specific activation of their targets. Such mechanisms may have arisen due to the ability of Hox TFs to recognise a variety of binding sequences.”

#3. The authors carried out multiple EMSAs to test putative Dfd-Exd sites. Interestingly, they did not find a clear correlation between the predicted relative binding affinity and the observed binding efficiency (eg page 7 Oligo C vs Oligo A). Overall, all sites predicted to be of low affinity were actually bound with “comparable high affinities”. With reference to what standard the authors can qualify the binding as with “high affinity”? That the correlation between predicted binding and observed binding is poor is one thing, qualifying the observed binding as of high affinity needs to be justified and objectified.

Our initial EMSAs experiments, done to characterise the potential Dfd-Exd sites present in *AP2x-377* (Fig. 4 and Supplementary Fig. 6) showed that Dfd-Exd complexes bind with similar efficiency to all predicted Dfd-Exd binding sites (Fig. 4h). However, as the reviewer correctly mentions, these results only indicate that the relative affinity of Dfd-Exd complexes for these sites is similar. Therefore, following the reviewer’s suggestion to better characterise the affinity of these binding sites, we determined the equilibrium dissociation constants (Kds) of Dfd-Exd complexes bound to the identified binding sites in *AP2x-377*. To have an accurate determination of the strength of these sites, we compare the calculated Kds and compared them with the equilibrium dissociation constant of a standard high affinity Dfd-Exd binding site. The standard we used was the consensus class 2 Dfd-Exd site as defined by high-throughput SELEX experiments (Slattery et al., 2011). According to this and other studies, this sequence is high affinity binding site for Dfd-Exd. According to the published literature, the definition of low-affinity binding sites is somewhat unclear. While it has been reported that low-affinity binding sites have a 1000-fold reduction of affinity as compared to consensus binding sequences, other studies reported that the Ubx low-affinity sites present in the *shavenbaby* enhancers have 5% of binding efficiency (20-fold less) as compared to the optimal Ubx-Exd binding sites as defined by SELEX experiments (Slattery et al., 2011).

To determine the affinity of the Dfd-Exd binding sites in *AP2x-377*, we determine the equilibrium dissociation constants for sites 10 and 11 present in domain 3, as these sites were required for the activation of *AP2x-377 in vivo* (Fig. 5e-e”).

As you may see in the new Supplementary Figures 7 and 8, under our experimental conditions Dfd-Exd complexes bound an oligo containing the canonical class 2 Dfd-Exd site with a Kd= 9.1 nM. Accordingly, the equilibrium dissociation constants determined for Dfd-Exd complexes bound to sites 10 (-AATACTAATCTA-) and site 11 (-GCACCTAATGAC-) showed that the Dfd-Exd bound with similar affinities to both sites (Kd= 16.8 nM) and only 1.8-fold smaller affinity of the consensus class 2 Dfd-Exd high affinity binding site.

We also determined the affinity of Dfd-Exd when we introduced mutations to convert sites 10 and 11 into class 2 consensus Dfd-Exd binding sites (site 10^{con}-TTGATTAATTAA-; site 11^{con}-TTGATTAATGAC-). As seen in Supplementary Fig. 7 and 8, optimization of these site resulted in a 2-2.5-fold increase in affinity (site 10^{con}, Kd=8.16 nM; site 11^{con}, Kd= 6.85 nM).

Therefore, these results confirm that the Dfd-Exd binding sites present in *AP2x-377*, although deviating from the class 2 Dfd-Exd high affinity binding site consensus (site 10) and in some cases from the core AYNNA Y Hox-Exd consensus sequence (site 11), are high-affinity binding sites, which are bound with a similar affinity than the consensus class 2 Dfd-Exd binding sites.

#4. Now, when the authors state that (end of section page 7) "...experiments identified multiple Dfd-Exd binding sites....all of which were bound by Dfd-Exd complexes with comparable high affinities", this is also not completely correct since, according to the oligos designed, all sites have not been tested individually. Indeed (cf Fig.3 and suppl Fig. 7), sites 3, 5, 7 or 11 have been analyzed (mutated) separately, while not sites 9 and 8.1/2/3, sites 2 and 4 or sites 1 and 6. The fact that some predicted sites overlap makes drawing conclusions obviously difficult about individual sites. But this calls for a comment while describing/discussing EMSAs.

We thank you the reviewer for the comment. We have modified this part of text to describe more accurately our characterization of the Dfd-Exd binding sites present in *AP2x-377* and at the same time to comment on the constraints that overlapping Dfd-Exd binding sites impose of a detailed study of these sites. At the end of this section it now reads:

"Overall, with exception of binding sites 5 and 6, all predicted Dfd-Exd sites in *AP2x-377* were bound by Dfd-Exd complexes. While we were able to study some of these sites individually (sites 1, 9, 10 and 11), the substantial overlap of other sites made it difficult to ascertain their individual contribution (sites 2/3/4 and 7/8). Regardless, and despite the predicted relative low-affinity, the identified Dfd-Exd sites were bound rather efficiently by Dfd-Exd complexes (Fig. 4h)."

#5. While "optimizing" core binding sites, the authors observed a modest increase in the binding of Dfd-Exd. This however has been observed while involving 120 ng of Dfd (Suppl Fig. 8 b). It is possible that a more differential binding would have been observed while adding lower amount of proteins. To conclude that optimizing the sequence has a modest effect on affinity requires testing a certain range of protein concentration to show that affinity/dissociation curves almost overlap. This is really important in particular since the authors highlight a discrepancy between the modest effect of optimizing the core (and flanking sequences) and the fact that the optimized sequences display a significantly different response (reporter, phenotype) according to low- and high-Dfd concentration territories. If the *AP2x-377* and *AP2x-377-D23H* enhancers are sensitive to Dfd levels (Fig.6), and if the optimized enhancer supports spatial mis-regulation in low-Dfd cells (and impairs maxillary structures development), I would suggest the authors to repeat EMSAs on wt versus optimized sequences with distinct Dfd amounts to possibly resolve this discrepancy.

As discussed in point #3, we performed a more detailed analysis of the Dfd-Exd binding sites present in *AP2x-377*. Following the reviewer's suggestion, we determined the equilibrium dissociation constants (Kds) for sites 10 and 11 present in domain 3 of *AP2x-377* (new Supplementary Figures 7 and 8). We previously had shown that both sites are required for the activation of *AP2x-377 in vivo* and optimisation of these sites resulted in the ectopic activation of *AP2x-377* in the more anterior-ventral regions of the maxillary segment. Therefore, we determined the affinity/dissociation curves of Dfd-Exd for these sites. To that end, we performed EMSAs with a wide range of concentrations of Dfd, from as low as 1.8 nM to a 59 nM (a 32-fold increase). From these experiments we were able to plot the affinity curves and calculate the respective Kds. We have now determined that in Dfd-Exd binds these sites with identical affinity (site 10: Kd=16.87 nM; site 11: Kd= 16.79 nM) and that optimisation of sites 10 and 11 to class 2 canonical Dfd-Exd sites results in a 2 and 2.5-fold increase respectively, values that are in the same order of magnitude as our first reported results. Overall, these experiments show that the Dfd-Exd binding sites in *AP2x-377* are high-affinity sites and that optimisation of these sites results in a small but significant increase of Dfd-Exd binding, resulting in ectopic activation of *AP2x-377* in the maxillary segment. These results reinforce our model in which activation of *AP2x-377* depends on a precise balance between the concentration of Dfd throughout the maxillary segment and the affinity of the Dfd-Exd binding sites in *AP2x-377* with a small increase in the affinity of Dfd-Exd sites resulting in the activation of the enhancer in cells of expression lower levels of Dfd.

#6. Related to this, how the proteins involved in EMSAs were quantified needs clarification. In the Methods section the authors indicate that proteins were quantified "using Fiji". Is this method reliably and reproducibly

quantitative? On some EMSAs (eg Figure 3e or Suppl. Fig. 7b), the amount of Dfd involved is indicated (38 ng, 76 ng, ...). To what extent are these values reliable, have they been determined according to reference samples of known concentration? Quantification of proteins requires clarification. This is important to support the conclusion that Dfd binds sites with “comparable high affinity” and that “optimizing the sequence has a modest effect on affinity”.

We thank the reviewer for calling our attention to these details. We have now introduced in the Materials and Methods section a more detailed description of how the proteins used to conduct the EMSAs experiments were quantified. In order to be precise in our quantification, we used samples of known concentration to calculate the concentration of our proteins. Overall, the methodology used is reliable and reproducible. In addition, as pointed out above, we have now complemented this quantification with Kd calculations, which are consistent with our previous quantifications.

#7 Comparing the expression patterns of the AP2x-377, -D3H and -D23H reporters, the authors indicate (page 9, cf Figure 5) that the -D3H and -D23H reporters display similar ectopic expression, supporting that Domain 3 of the AP2x enhancer is crucial. While looking at Figure 5 c', d' and f', it seems that the ectopic expression provided by D23H is more extended and, visually, the pattern provided by -D3H appears more similar to the wt AP2x-377 than to -D23H.

Regarding this point, we respectively disagree with the reviewer's comments. Of the two conserved domains present in AP2x-377, domain 3 is the only one required for the activity of AP2x-377. Mutation of the Dfd-Exd binding sites in domain 3 result in the inactivation of the enhancer while the destruction of the Dfd-Exd sites in domain 2 had no effect in the activity of the enhancer. Supporting these results, optimisation of Dfd-Exd sites in domain 2 showed almost no changes in the activity of the AP-2 enhancer, with its activation pattern being indistinguishable from the wild-type enhancer. However, the optimisation of the Dfd-Exd sites of domain 3 resulted in a clear and distinct ectopic activation of AP2x-377-D3H in the anterior-ventral regions of the maxillary segment, broader than the activation pattern of the wild-type AP2x-377 enhancer. A similar result is observed with AP2x-377-D23H, where again we observed an ectopic expression similar to the one observed for AP2x-377-D3H and clearly broader than the wild-type enhancer AP2x-377.

To make these observations clearer and easy to visualize, we introduced the following changes in Fig. 6c-g". We delineated the posterior border cells of the maxillary segment where Dfd is highly expressed (dashed yellow line). With the exception of the activity displayed in a medial stripe across the maxillary segment, the activation of AP2x-377 occurs in the posterior border cells of the maxillary segment expressing high-levels of Dfd (Fig. 6c-c"). Therefore, this domain can be used as a reference to observe the ectopic activation of both AP2x-377-D3H and AP2x-377-D23H. It is now easier to observe that for AP2x-377-D3H, AP2x-377-D23H and AP2x-377-D2mt-D3H, ectopic activation of these enhancers takes place in cells located anteriorly and ventrally to the labelled Dfd domain (Fig.6d', f' and g'). In the same way, it is now clear that AP2x-377-D2H shows an activation pattern indistinguishable from the wild-type AP2x-377 enhancer (Fig. 6c' and e').

#8. Along the same line, in the discussion section (page 13, §2) the authors state that "...the AP2x-377-D23H enhancer is more sensitive to Dfd levels than AP2x-377." This cannot be affirmed based on the data since the wt and -D23H enhancers might be sensitive to different ranges of Dfd concentrations, which has not been addressed.

As discussed in point #5, we have now calculated the equilibrium dissociation constants (Kds) for both sites 10 and 11 present in domain 3 of AP2x-377. We can now show that both sites are Dfd-Exd high-affinity sites. We have also calculated the Kds for the optimal versions of both these sites and found that converting these sites to canonical class 2 Dfd-Exd binding sites leads to a 2 and a 2.5-fold increase in affinity for sites 10 and 11 respectively. Thus, these results indicate that the mutations introduced in AP2x-377-D23H to optimise the Dfd-Exd binding sites increase the affinity of Dfd-Exd complexes towards the binding sites present in the enhancer and as a result these enhancers are now more sensitive to concentrations of Dfd. These results

support our observations that AP2x-377-D23H is more sensitive to the concentration of Dfd than the wild type enhancer AP2x-377. As seen in Supplementary Fig. 10, when overexpressing Dfd in the embryonic ectoderm AP2x-377 displays an ectopic activation restricted to the anterior cephalic regions, specifically in the antennal segment (Supplementary Fig. 10a-a" and b-b"). The same was observed when Dfd was over-expressed in AP2x-377-D23H embryos (Supplementary Fig. 10c-c", d-d"). However, AP2x-377-D23H was also active in more posterior segment, from the labial to the third thoracic segment (Supplementary Fig. 10c-c", d-d"). Therefore, these results together with the newly calculated equilibrium dissociation constants for both wild-type and optimised Dfd-Exd binding sites of domain 3 indicate that optimisation of the Dfd-Exd sites results in AP2x-377-D23H being more sensitive to Dfd levels and that the activity of the AP-2 enhancer across the maxillary enhancer depend on the balance between the affinity of the Dd-Exd binding sites and the levels of Dfd expressed throughout the maxillary segment. In addition, we have now also performed ChIP experiments in embryos carrying the different transgenes, which shows that Dfd binding is stronger to the AP-2 enhancer harbouring the high affinity Dfd-Exd sites.

MINOR POINTS:

1. In the abstract, the "TF" abbreviation is not defined

The appropriate definition of TF was introduced in the first sentence of the abstract.

2. Introduction, page 3, the authors indicate "...Hox TFs share the same DNA-binding domain, the homeodomain". I would indicate "highly similar" or "extremely conserved" rather than "the same".

Following the reviewer's suggestion, we introduced the following modification. Where it read, "*This latter question is particularly important as Hox TFs share the same DNA-binding domain, the homeodomain*", it now reads

"This latter question is particularly important as Hox TFs share a highly conserved DNA-binding domain, the homeodomain."

3. While identifying the sequence determinants of the AP-2 enhancer activity, the authors conclude that the "AP2x-377 represents the minimal enhancer for Dfd-dependent activation". This is not precisely correct, AP2x-377 represents the minimal enhancer tested providing Dfd-dependent activation, thus it contains the minimal information but does not define the minimal enhancer. More sequence dissection could lead to a more refined definition of the necessary regulatory elements defining the enhancer. So, I would suggest to rephrase this sentence "In sum, these results revealed that... represents the minimal enhancer...." on page 6.

We introduce modifications in the mentioned sentence to accommodate with the reviewer's comments. Where it before read "*In sum, these results revealed that the AP2x-377 fragment represents the minimal enhancer for Dfd-dependent AP-2 activation in the maxillary segment.*" it now reads "**In sum, these results revealed that the AP2x-377 fragment contains the minimal information required for Dfd-dependent AP-2 activation in the maxillary segment.**". However, our molecular dissection of AP2x strongly suggests that AP2x-377 is the minimal enhancer capable of driving expression of AP-2 in the maxillary segment. As the reviewer may see in Fig. 2, further deletion of AP2x-377 at either 5' or 3' with the removal of domains 2 and 3 respectively, resulted in inactivation of the enhancers at stage 10 and reduced activity at later stages (stage 13).

4. Looking at Figure 2 presenting reporter expression patterns, Fig. 2 b and d seem to show that the AP2x and AP2x-377 enhancers do not provide reporter activity in identical territories. The AP2x-377 enhancer seems to drive ectopic expression at st11. This appeals for a comment in the text, pages 5-6. Similarly, the AP2x-230 and -268 enhancers still promote some signal (Fig. 2f, g) while the authors indicate that the deletions made to generate these -230 and -268 enhancers "resulted in loss of enhancer activity in earlier stages". Does the signal visible on Fig. 2f and g correspond to background ectopic signal?

We thank the reviewer for pointing our attention to this detail. Indeed, the *AP2x-377* enhancer drives an ectopic expression in the procephalic/antennal lobe and the same is observed for all subsequent deletions of this enhancer. To clarify these observations, we modified the text to add a more detailed description of these results. However, despite this ectopic activation in the antennal region, the *AP2x-377* is activated in an identical pattern as the original *AP2x* enhancer in the maxillary segment in both stages 11 and 13 (Supplementary Fig. 4a-b”).

5. While describing EMSAs, the authors should provide more details about quantification. Graphs appearing on Fig.3 c, f, g and Suppl. Fig.7 c, f, g present “Binding %”, but what defines the 100%, how the quantification is normalized should be indicated.

We introduced in the Materials sections a more detailed account of how the EMSAs were quantified. The EMSAs were analyzed with Fiji and binding affinities quantified as the percentage of bound oligonucleotide over the total amount of oligonucleotide (bound and unbound) in the reaction.

6. Next on Fig.3 b, e and Suppl Fig. 7b, e, EMSA gels are presented with two frames each, a wide, top panel showing non-retarded and retarded bands, bottom panels appearing as the non-retarded bands but with lower exposure, is that correct? A better description of what is presented should be provided (legend, maybe arrows or arrowheads to identify bands...).

The reviewer’s observation is correct. The mentioned bottom panels are the non-retarded bands (unbound oligo) with a lower exposure. To simplify the image, these panels have been removed from these Figures which are now Fig.4c, f and Supplementary Fig. 6c, f.

7. In the discussion section, the authors indicate (page 12, §3): “Additional cofactors are required to interact specifically with the Dfd-Exd complexes bound to domain 3. This is supported by our observation that the non-conserved region...is required”. Although it is indeed highly probable that additional factors may be required to interact with Dfd-Exd onto domain 3, this sentence is somehow too affirmative since non-conserved sequences in a regulatory sequence can be influential as such by facilitating the cooperative interaction between proteins, eg thanks to sequence-driven conformation of the DNA (DNA bending for example, opening of grooves, etc...).

We thank the reviewer for its comment. We modified this part of the discussion to accommodate the reviewer’s suggestion.

Reviewer #2: thought the study is highly significant for our understanding how gene expression is controlled in vivo. However, this reviewer also thought that the manuscript needs some improvements before it can be published in Nat Comm. We addressed the comments of this reviewer in the following way:

MAJOR POINTS:

#1. I have not found a clear and straightforward explanation of how the authors define high affinity sites. Are those based on in vitro studies? Clearly, their experiments show that low affinity sites are high affinity in vitro as well. What is the supporting evidence for high affinity sites? This is crucial to distinguish high and low affinity sites. Otherwise it risks becoming a semantic exercise...

High-affinity binding sites are defined as sites to which TFs display an optimal binding, i.e, consensus sequences. The value of the affinity of a particular TF to a specific binding site is obtained by calculating the

equilibrium dissociation constant (K_d) associated to that particular binding event. Therefore, the K_d associated to the binding of a TF to its consensus binding sequence defines this value and the order of magnitude that is expected for a high-affinity site. In the case of the Hox-Exd complexes, SELEX analysis using the different HOX-Exd complexes have defined optimal binding sites for the different complexes. As we explain in the introduction, these experiments characterised groups of binding sites that are preferentially recognized by the different Hox-Exd complexes (Slattery et al., 2011). As a result, Hox complexes are organized in three classes according to their binding site preferences. Class 1 binding sites, with a nTGATTGATnnn core sequence, are bound preferentially by Labial (Lb) and Proboscipedia (Pb), while class 2 sequences with a nTGATTAATnnn core sequence are preferred targets of Deformed (Dfd) and Sex comb reduced (Scr). Class 3 Hox-Exd complexes are composed of Antennapedia (Antp), Ultrabithorax (Ubx), Abdominal-A (Abd-A) and Abdominal B (Abd-B) and show a higher affinity towards nTGATTTATnnn core sequences. These results provided framework to understand Hox-Exd high-affinity sites, and it is described like that in the Introduction.

Low-affinity sites affinity binding sites have been described as sequences that do not resemble the Hox-Exd consensus binding sequence and are bound by Hox-Exd complexes with a lower affinity. In the case of the shavenbaby Ubx-regulated enhancers, the low-affinity binding sites identified in these regulatory elements are bound by Ubx-Exd with 20-fold less affinity as compared to the class 3 Ubx-Exd High-affinity binding sites identified in the SELEX experiments. Recently in a review by Mann et al 2019, the authors describe low-affinity sites as having a 1000-fold less affinity as compared to the consensus sequences.

We now have detailed our characterisation of the Dfd-Exd binding sites present in AP2x-377. In our earlier EMSA experiments we determined that all identified Dfd-Exd binding sites were bound with similar efficiency by Dfd-Exd complexes. Therefore, we have determined the equilibrium dissociation constants (K_d) to have a true value of the affinity of the Dfd-Exd sites. We focused on the Dfd-Exd binding sites 10 and 11 present in domain 3 as these are required for in vivo activation of AP2x-377 (Figure 5e-e''). At the same time, we determined also the K_d for the binding of Dfd-Exd complexes to an oligo containing a high-affinity class 2 Dfd-Exd binding site. From these experiments we obtained K_d values of 16.87 nM and 16.79 nM for sites 10 and 11 respectively, while Dfd-Exd complexes bound to a class 2 Dfd-Exd high-affinity site display a value of $K_d=9.12$ nM (Supplementary Fig. 7 and 8). This indicates that Dfd-Exd complexes bound to sites 10 and 11 of AP2x-377 with 1.8-fold lower affinity than to a class 2 Dfd-Exd high-affinity site. In addition, we determined the changes in affinity of both sites 10 and 11 when we introduce mutations to convert these sites to class 2 Dfd-Exd high-affinity sites. Overall, we saw a 2-2.5-fold increase in affinity (Supplementary Fig. 7 and 8), a value of the same order of magnitude as determined by our previous calculations. Thus, in our experimental conditions, we determined that the Dfd-Exd sites present in AP2x-377 are high-affinity sites with affinities close to the ones displayed by class 2 Dfd-Exd high-affinity sites.

#2. The differences in Dfd in vitro binding after optimization of the sites are quite small as the authors have noted. Have they considered the possibility that mutagenesis of these sites rather than optimizing Dfd binding in vivo can decrease binding of a repressor (which is competing with Dfd for accessing the AP2 enhancer)?

The reviewer's hypothesis is certainly pertinent. The existence of a repressor is difficult to prove and therefore we can't certainly not simply dismiss such possibility. However, all our data strongly supports the conclusion that the activation of AP2-377 in the maxillary segment results from a balance between the affinity of the Dfd-Exd binding sites present in the AP-2 enhancer and the concentration levels of Dfd across the maxillary segment.

As mentioned in the point #1, our *in vitro* data clearly shows a 2-2.5-fold increase in affinity of Dfd-Exd complexes when the binding sites are converted in class 2 consensus Dfd-Exd high-affinity sites (Supplementary Fig. 7 and 8). We have now performed Chip experiments to determine the binding of Dfd to AP-2 enhancer *in vivo* when we optimise Dfd-Exd binding sites (Supplementary Fig.8d). As we saw with the EMSAs experiments, converting the Dfd-Exd sites to class 2 consensus sites modestly increased the binding of Dfd *in vivo*.

We further tested our model by modifying the levels of Dfd in the maxillary segment. As you may see in the new Figure 7, we used the UAS/Gal4 system to increase the levels of Dfd in the maxillary segment. More specifically we use the *AP2-377-D23H-Gal4* driver to increase the levels of Dfd in the anterior region of the maxillary segment (Figure 7e-e''). The increase of Dfd in this region resulted in the ectopic activation of *AP2x-377* (Figure 7a-d''). This result supports and complements our initial experiments showing that the increase affinity of Dfd-Exd binding sites in *AP-2* enhancers resulted in ectopic activation in the anterior region of the maxillary segment. Furthermore, using *ptc-Gal4* as a driver, we performed RNAi of Dfd in the anterior part of the maxillary segment to determine if the decrease in levels of Dfd could reduce the activity of *AP2x-377-D23H* (Fig.7f, j-j''). We were able to reduce Dfd levels in the anterior cells of the maxillary segment resulting in a reduction of ectopic activity of *AP2x-377-D23H* (Figure 7g-i''). All expression changes are quantified as shown in Figure 7m.

Thus, these results show that either small variations in Dfd-Exd binding affinity or levels in the Dfd result in a loss of domain-specific activation of *AP2x-377* in the maxillary segment. Overall, the new data strongly supports a model where activation of *AP2x-377* in the maxillary segment is the result of a balance between the affinity of the Dfd-Exd binding sites present in the *AP-2* enhancer and the concentration levels of Dfd across the maxillary segment.

MINOR POINTS:

1. Figure 2: what are 1, 2 and 3 in blue and what are the yellow areas? I only understood that by looking at the corresponding supplementary figure

We thank you the reviewer for bringing that to our attention. We have now reorganized the paper figures, so that in critical figures the organization of the *AP2x-377* enhancer is included (Figures 3, 4, 5, 6). In particular, we moved a previous supplementary figure into the main figure section (now Figure 3), in which the organization of the *AP2x-377* enhancer is shown with all the predicted Dfd-Exd binding sites.

2. Figure 3:

2.1. it was rather difficult (and time consuming) to piece together all the information required to fully interpret the results in this figure and their conclusion. Part of the information is in the result text and part in the supplementary figure. It was unclear how these sites were selected to be tested in EMSA, and where they are in the enhancer. The authors should reorganise the figure by adding a drawing of the enhancer showing the location of the sites and merging with panels from supplementary figure 5 to show the sequence of the sites.

We reorganised the figures and supplementary figures to clarify the information and interpretation. In this case, the previous Figure 3 and Supplementary Figure 5 are now Figures 4 and 3, respectively. In addition, as the reviewer requested, a drawing of *AP2x-377* was added in the new Figure 4 as to display the position of the identified Dfd-Exd sites. As for how the sites were selected to be tested by EMSA, we provided additional information in the Results section which we hope will clarify the methodology used.

2.2. The oligos used in EMSA should be renamed to reflect the nomenclature of the sites.

We re-labeled the oligos and the Dfd-Exd sites.

2.3. The authors should test a canonical Dfd motif to quantify the relative binding of Dfd to 'low affinity' sites

As mentioned in point #1, we have now determined the equilibrium dissociation constants (K_d s) for Dfd-Exd complexes binding to a class 2 Dfd-Exd high-affinity site (identified in previous SELEX experiments (Slattery et al., 2011) and to the Dfd-Exd sites 10 and 11 in domain 3 of *AP2x-377*. As the reviewer can see in the new Supplementary Figures 7 and 8, Dfd-Exd complexes bound a class 2 high-affinity binding site with a $K_d = 9.12$

nM while for AP2x-377 sites 10 and 11, the determined Kds were 16.87 nM and 16.79 nM respectively. In other words, Dfd-Exd complexes bound a class 2 high-affinity site with an affinity only 1.8-fold higher than to sites 10 and 11 of AP2x-377. We also determined the equilibrium dissociation constants for sites 10 and 11 when mutations were inserted to convert these sites in to class 2 Dfd-Exd high-affinity sites. As seen in Supplementary Figures 7 and 8, Dfd-Exd bound optimized sites 10 and 11 with a Kd of 8.16 nM and 6.65 nM, respectively, showing that the optimization of these sites increased the affinity of Dfd-Exd complexes in 2-2.5-fold, a similar order of magnitude seen in the previous experiment. Therefore, these results support and strength our description of the Dfd-Exd binding sites present in AP2x-377 are high-affinity sites.

2.4. I assumed the probe was at the bottom of the main area outlined, but there is a smaller section outlined at the bottom of each gel - what is the band contained here? And what are the numbers underneath? I did not find any reference to those numbers in the text – if so they should be removed.

The smaller section at the bottom of each gel showed lower exposures of the unbound oligo. To clarify the image, these sections as well as the numbers underneath the several gels were now removed.

2.5. There is no description of how binding quantification were made.

More details were added to the Materials section to provide a full description of how the binding quantification was made.

3. Supplemental Figure 5:

3.1. The graph shows different affinities – how does the search tool classify those identified as low affinity binding sites? If it defines sites as low affinity based on their distance from a high affinity site, does the search tool definition of high affinity site coincide with the one of the authors?

The NRLB algorithm used in this study is an equilibrium thermodynamics model of protein-DNA interaction. It has been developed to improve the analysis of SELEX data (Rastogi et al PNAS, 2018). It predicts the existence of TF binding sites and their relative binding affinity ($K_{a, rel}$). Core to the determination of the relative binding affinity, is the determination of the relative binding energy $\Delta\Delta G(S) = \Delta G(S) - \Delta G(S_0)$ where S_0 is a fixed reference, chosen to be the highest-affinity sequence, i.e, consensus sequence. According to Rastogi et al., several features are taken into account to determine the relative binding affinity including all possible mononucleotides substitutions (mononucleotide model) as well as dependencies among pairs of adjacent nucleotides (dinucleotide model) which has the potential to improve prediction as it takes into account the effects of variation in DNA shape on binding. Thus, this algorithm does not provide a true measure of TF binding affinity but rather a relative affinity, having as a reference a high-affinity sequence previously established.

To compare the predicted binding affinities between the binding sites present in the AP2x-377 enhancer and a Dfd-Exd high-affinity site, we now artificially introduced a high-affinity Dfd-Exd sequence in the non-conserved region between domain 2 and 3 and ran the NRLB algorithm again using this sequence as input. The result is displayed in Supplementary Fig. 5c and shows that the canonical Dfd-Exd high affinity site is predicted to have more than 100fold higher affinity for the Dfd-Exd complexes than the sites present in the enhancer. So, according to the NRLB algorithm, the sites in the AP2x-377 enhancer are of low-affinity in relation to a known high-affinity site.

To accurately measure the affinity of Dfd-Exd complexes for the different binding sites in AP2x-377, we now made experiments to determine the corresponding equilibrium dissociation constants (Kds). As discussed in point 2.3 of the specific comments, the calculation of the Kds for the different binding sites present in AP2x-377 as well as for class 2 Dfd-Exd high-affinity sequences showed that Dfd-Exd complexes bound the NRLB predicted low-affinity binding sites with an affinity only 1.8-fold lower as that exhibited for class 2 Dfd-Exd high-affinity sequences. Thus, although the NRLB has predicted correctly the existence of Dfd-Exd binding sites present in AP2x-377, it failed to provide a relative affinity that accurately reflected the experimentally

determined equilibrium dissociation constants. We don't fully understand the reasons for such discrepancy. The NRLB algorithm has shown to efficiently determine binding sites for several TFs including Ubx-Exd low-affinity sites present in the E3N and 7H *shavenbaby* enhancers (Rastogi et al PNAS, 2018). A possible explanation could lie on how the algorithm has been developed to analyse SELEX data. In these experiments, oligos are used with a 12-nt central core region where all nucleotide permutations are included, and flanked by invariant sequences. These flanking sequences might account for the discrepancy in binding affinities. It may be possible that these regions might contribute to the overall binding of Dfd-Exd. We have included these hypotheses now in the Discussion.

3.2. What does class 2DE in b means? And what are the yellow and brown highlights?

The images were rearranged to improve the flow and taking of the paper. Supplementary Figure 5 is now Figure 3. In this figure "Class 2 DE" refers to an optimal high-affinity class 2 Dfd-Exd binding site. Such information was now added to the figure legend.

4. Figure 4: Can the expected phenotype be described for the non-specialist reader? Is it the number/shape of cirri that is rescued and/or is defective?

This figure is now Figure 5. We made modifications to the text to improve the description of the phenotype. Overall, *AP-2¹⁵* mutant larvae fail to develop any ventral cirri. To evaluate the effect of the mutations in domain 2 or domain 3 in the activity of *AP-2* enhancer, we used these enhancers as drivers to express *AP-2* in *AP-2¹⁵* mutant larvae. According to our genetic experiment setup, only 50% of *AP-2¹⁵* mutant larvae should express *AP-2*. As seen in Figure 5g-i and Table 2, only the AP2x-377-D3mt enhancer failed to properly rescue *AP-2¹⁵* mutant larvae (27.8%)

5. Figure 5: a. what does H mean? I will put this in the legends. Why is the strategy to assess enhancer activity different here? Instead of being upstream of a reporter (as in previous figures) the enhancer is placed upstream of GAL4 and the readout is obtained by crossing with UAS-lacZ strain.

This Figure is now Figure 6. "H" indicates the Dfd-Exd sites that have been converted to optimal class 2 Dfd-Exd high-affinity sites. This information was now properly added to the figure legend. As for why the strategy to assess the enhancer activity was changed, the reason is simply practical. The primary goal of this study has been to determine *in vivo* effects in changes of affinity of Dfd-Exd sites and their contribution to the morphogenesis of the specific head structures, the ventral cirri. To this end, we decided to perform the rescue of *AP-2¹⁵* mutants by directing *AP-2* expression under the control of different *AP2x-377* mutant enhancers. Therefore, we constructed Gal4 lines with all *AP2x-377* mutant enhancers that we generated. Due to the number of transgenic lines that would have been required to obtain all the required GAL4 4 lines plus the equivalent reporter lines driving an appropriate reporter, we opted to generate only the Gal4 lines and take the advantage of the UAS-GAL4 system to perform both the rescue by crossing with *UAS-AP-2* and the characterization of the activity of the different *AP2x-377* mutant enhancers by crossing with *UAS-LacZ*.

6. Figure 6: The observation that optimization of Dfd sites increases the response of the enhancer to Dfd, but does not change enhancer specificity is very interesting. The authors should back up this information showing how Scr binds in vitro to enhancer exemplar low affinity sites and their optimised counterparts.

We have now introduced new experiments that tackle this point. Our initial experiments have shown that converting the Dfd-Exd sites present in *AP2x-377* to class 2 high-affinity sites (*AP2x-377-D3H* and *AP2x-377-D23H*) resulted in the ectopic activation of the enhancer within the maxillary segment (Figure 6d-d", f-f" and Supplementary Figure 9). Although class 2 Hox-Exd binding sites are also preferentially bound by Scr-Exd complexes (Slattery et al., 2011), no activation of *AP2x-377-D23H* was observed in the labial segment where Scr is expressed (Supplementary Figure 10c-c"). Ectopic expression of UAS-Dfd using the ectodermal driver 69B-Gal4 resulted in the ectopic activation of *AP2x-377-D23H* outside the maxillary segment, from the labial

to the third thoracic segment (Supplementary Figure 10c-d"). The activation in the labial segment suggests that despite the *AP-2* enhancer is not active in this segment, the labial segment contains all the factors required for the activation of the *AP-2* enhancer. To understand why *Scr* was unable to activate *AP2x-377*, we performed EMSA experiments and determine the equilibrium dissociation constants (*K*_d) for *Scr*-Exd complexes bound to site 11 of *AP2x-377*. As shown in the new Supplementary Figures 7a, b and 11a, b, EMSAs were performed using oligo E-wt that contains the wild-type site 11. *Scr*-Exd complexes displayed a 7-fold lower affinity when compared to *Dfd*-Exd sites (*Scr*-Exd, *K*_d= 120 nM; *Dfd*-Exd, 16.8 nM). Interestingly, when we performed the same experiment using oligo E-11H that contains mutations to convert site 11 into a class 2 Hox-Exd high-affinity binding site, *Scr*-Exd display a further 4.6-fold reduction in the affinity to the optimal site (Supplementary Figure 11a, c). Overall, our data suggest that the Hox cofactors required for the activation of *AP2x-377* are present in both Maxillary and Labial segments and that specific activation of *AP2x-377* in the maxillary segment seems to rely on the differential affinity with which *Dfd*-Exd and *Scr*-Exd complexes bind Hox-Exd sites present in *AP2x-377*.

7. Figure 7/8: I suggest merging these two figures (possibly with the help of a supplementary figure) and move them immediately after Fig. 5 (Fig 6 interrupts the logic of the flow).

We thank the reviewer for this comment. We follow the reviewer's suggestion and merged Figures 7 and 8 in what is now Figure 8. Figure 6 was also changed and is now Supplementary Figure 9. We hope that these modifications improve the flow of the paper.

8. P4: This sentence is unclear: "This simplistic view of binding site affinity and Hox specificity certainly holds true for posterior Hox proteins, as the activation of their target genes needs to take into account the promiscuous binding of the anterior Hox-Exd complexes that imposes constrictions to the posterior Hox proteins".

To clarify the aim of this study this sentence has been removed and replaced by "**The existence of Hox-Exd low-affinity binding sites provides an explanation of how posterior Hox-Exd complexes activate their specific target genes in a tissue-specific manner while preventing activation by anterior Hox-Exd complexes, capable of binding a wider range of binding sites.**". We hope this will help framing the proposed aim of this study.

Reviewer #3: appreciated that this study characterized *Dfd* binding to the *AP2x* enhancer *in vitro* and correlated this with enhancer function *in vivo*. The reviewer had several concerns that we addressed in the following way:

MAJOR POINTS:

#1. The fact that many enhancers contain suboptimal binding sites and that optimization of such enhancers leads to abnormal function is well established.

It is true that many enhancers have been described as having suboptimal binding sites that are required for their specific spatial-temporal activation. However, it is not the aim of this work to study suboptimal binding sites and their role in gene expression but rather to understand *in vivo*, the molecular mechanisms used by anterior Hox TFs to ensure the spatial-temporal activation of its target genes. In addition, our study goes beyond a simple description that suboptimal binding sites are critical for precise target gene regulation, but most importantly shows that their usage is critical for segment morphogenesis, as we test the morphological consequences of mis-regulation induced by optimal binding sites. This was also highly appreciated in particular by Reviewer 2 but also by Reviewer 1.

Anterior Hox TFs have been shown to preferentially bind class 2 Hox-Exd high-affinity sites. However, these Hox TFs are also able to bind class 3 Hox-Exd high-affinity sites which are preferentially bound by posterior

Hox TFs. It has been shown that the existence of Hox-Exd low-affinity binding sites in CRMs regulated by posterior Hox TFs provides an explanation on how these Hox TFs activate their target genes in a tissue-specific manner while preventing activation by anterior Hox-Exd complexes (Crocker et al., 2015). This is in agreement with the general view that low-affinity binding sites are features of enhancers that provide specific spatial-temporal regulation of transcription or that are regulated by specific TFs. Consequently, high-affinity binding sites seem to be employed when neither of these conditions seem to be required.

However, the promiscuity of the anterior Hox TFs raises the question of how these TFs factors are able to specifically activate their target genes and what is the role of affinity in this activation. We addressed this issue, by studied the embryonic maxillary enhancer *AP2x* of the *Drosophila* gene *AP-2/tfAP-2*. This enhancer directs the expression of the TF encoding gene *AP-2* in a specific domain of the maxillary segment, under the control of *Dfd*. *AP2x* lacks class 2 canonical *Dfd-Exd* sites, but contains instead several predicted low-affinity/non-canonical *Dfd-Exd* sites. This makes the *AP2x* enhancer an excellent model to address Hox-Exd binding site affinity and its role in Hox specificity.

We believe that the results from this study greatly contribute to our understanding of the molecular mechanisms employed by Hox TFs to achieve specificity. We have shown that:

- the anterior Hox TF Deformed is capable of binding with high-affinity to a wider range of binding sites than initially showed. In fact, based on previous published studies, these binding sites were predicted to be low-affinity binding sites as some of these sites fail to match either class 2 Hox-Exd high-affinity sites (described as preferentially bound by anterior Hox TFs) or the AYNAY core consensus of Hox-Exd binding sites.
- It shows that high-affinity sites can be used to achieve tissue, domain and temporal specific-activation of Hox target genes as *Dfd* specifically regulates the activation of its target gene *AP-2* in a specific subset of cells in the Maxillary segment.
- It shows that the *Dfd-Exd* binding sites in the *AP-2* enhancer, despite being high-affinity sites, are still under tight constraints to ensure region-specific regulation in a segment, as small increases in the affinity of these sites translates in loss of spatial and temporal regulation of the *AP-2* enhancer.
- The domain-specific activation of the *AP-2* enhancer in the maxillary segment relies on a balance between the affinity of the *Dfd-Exd* binding sites and the concentration levels of *Dfd* across the maxillary segment (which we now show in the additional Figure 7).
- the fine-tuned affinity of *Dfd-Exd* binding sites present in the *AP-2* maxillary enhancer is essential for *Dfd* to coordinate development of the other maxillary structures, such as the dorsal cirri and the mouth hooks.

#2. I do not believe that one can use data from a single enhancer and a single Hox TF to generalize 'a cis regulator code for anterior Hox specificity' as indicated by the title. To do so, it would be necessary to first look more broadly at high/low binding sites for other anterior Hox genes.

We agree with the reviewer's comment regarding the general view portrayed by the title and the lack of data supporting such statement. It is not the intention of this study to generalise its findings. The aim of this study is to understand how in vivo, anterior Hox TFs, which show a more promiscuous binding than posterior Hox TFs, are able to regulate the tissue-specific activation of its targets. Therefore, the title has been changed to "**A fine-tuned balance of transcription factor levels and binding site affinities determines specificity of the anterior Hox protein Deformed**", which we believe conveys more accurately the aim and conclusions of our work.

#3.

- The definition of 'low' and 'high' affinity sites is unclear. The use of the term 'low affinity' sites sometimes seems based on predictions in the literature and sometimes on their own EMSA data, which causes confusion. In fact, since the binding sites predicted to be low affinity (based on literature and NRLB) actually bind *Dfd* with high affinity, doesn't this simply suggest that the algorithm is faulty? To continue to refer to such sites as 'low affinity' seems disingenuous. It is also unclear how the authors define high versus low affinity in their EMSA

assay. There seems to be no absolute control included, instead this seems to be defined in relative terms. For instance, on what basis is Dfd binding to the AP2x elements defined as 'high affinity'. Is this a post-hoc conclusion based on the fact that Dfd binds the suboptimal and optimal sites with the same affinity?

The issues raised by Reviewer #3 are similar the ones raised by Reviewer #1 (point 3) and Reviewer #2 (point 1):

High-affinity binding sites are defined as sites to which TFs display an optimal binding, i.e, consensus sequences. The value of the affinity of a particular TF to a specific binding site is obtained by calculating the equilibrium dissociation constant (Kds) associated to that particular binding event. Therefore, the Kd associated to the binding of a TF to its consensus binding sequence setup the value and the order of magnitude that is expected for a high-affinity site. In the case of the Hox-Exd complexes, SELEX analysis using the different HOX-Exd complexes have defined optimal binding sites for the different complexes. As we explain in the introduction, these experiments characterised groups of binding sites that are preferentially recognized by the different Hox-Exd complexes (Slattery et al., 2011). As a result, Hox complexes are organized in three classes according to their binding site preferences. Class 1 binding sites, with a nTGATTGATnnn core sequence, are bound preferentially by Labial (Lb) and Proboscipedia (Pb), while class 2 sequences with a nTGATTAATnnn core sequence are preferred targets of Deformed (Dfd) and Sex comb reduced (Scr). Class 3 Hox-Exd complexes are composed of Antennapedia (Antp), Ultrabithorax (Ubx), Abdominal-A (Abd-A) and Abdominal B (Abd-B) and show a higher affinity towards nTGATTTATnnn core sequences. These results provided framework to understand Hox-Exd high-affinity sites.

Low-affinity sites affinity binding sites have been described as sequences that do not resemble the Hox-Exd consensus binding sequence and are bound by Hox-Exd complexes with a lower affinity. In the case of the shavenbaby Ubx-regulated enhancers, the low-affinity binding sites identified in these regulatory elements are bound by Ubx-Exd with 20-fold less affinity as compared to the class 3 Ubx-Exd High-affinity binding sites identified in the SELEX experiments. Recently in a review by Mann et al 2019, the authors describe low-affinity sites as having a 1000-fold less affinity as compared to the consensus sequences.

To determine the affinity of the Dfd-Exd binding sites in AP2x-377, we determine the equilibrium dissociation constants for sites 10 and 11 present in domain 3 as these sites were required for the activation of AP2x-377 *in vivo* (Fig. 5e-e"). As you may see in the new Supplementary Figures 7 and 8, under our experimental conditions Dfd-Exd complexes bound an oligo containing the canonical class 2 Dfd-Exd site with a Kd= 9.1 nM. Accordingly, the equilibrium dissociation constants determined for Dfd-Exd complexes bound to sites 10 (-AATACTAATCTA-) and site 11 (-GCACCTAATGAC-) showed that the Dfd-Exd bound identical affinities to both sites (Kd= 16.8 nM) and only 1.8-fold smaller affinity of the consensus class 2 Dfd-Exd High affinity binding site. We also determined the affinity of Dfd-Exd when we introduced mutations to convert sites 10 and 11 into class 2 consensus Dfd-Exd binding sites (site 10 ^{con}-TTGATTAATTAA-; site 11 ^{con}-TTGATTAATGAC-). As seen in Supplementary Fig. 7 and 8, optimization of these site resulted in a 2-2.5-fold increase in affinity (site 10 ^{con}, Kd=8.16 nM; site 11 ^{con}, Kd= 6.85 nM).

Therefore, these results confirm that the Dfd-Exd binding sites present in AP2x-377, although deviating from the class 2 Dfd-Exd High affinity binding site consensus (site 10) and in some cases from the core AYNNA Y Hox-Exd consensus sequence (site 11), are high-affinity binding sites bound by an overall affinity similar to that displayed by the consensus class 2 Dfd-Exd binding sites.

Concerning the NRLB algorithm: we do not believe that the algorithm is per se faulty, as it has been used in other studies (Sanchez-Higueras et al., 2019), where predictions were consistent with expected affinities. In addition, this algorithm proved very useful in identifying the different binding sites in the AP2x-377 enhancer.

To explain, the NRLB algorithm used in this study is an equilibrium thermodynamics model of protein-DNA interaction. It has been developed to improve the analysis of SELEX data (Rastogi et al PNAS, 2018). It predicts the existence of TF binding sites and their relative binding affinity ($K_{a, rel}$). Core to the determination of the relative binding affinity, is the determination of the relative binding energy $\Delta\Delta G(S) = \Delta G(S) - \Delta G(S_0)$ where S_0 is a fixed reference, chosen to be the highest-affinity sequence, i.e, consensus sequence. According to Rastogi et al., several features are taken into account to determine the relative binding affinity including all

possible mononucleotides substitutions (mononucleotide model) as well as dependencies among pairs of adjacent nucleotides (dinucleotide model) which has the potential to improve prediction as it takes into account the effects of variation in DNA shape on binding. Thus, this algorithm does not provide a true measure of TF binding affinity but rather a relative affinity, having as a reference a high-affinity sequence previously established.

To compare the predicted binding affinities between the binding sites present in the AP2x-377 enhancer and a Dfd-Exd high-affinity site, we now artificially introduced a high-affinity Dfd-Exd sequence in the non-conserved region between domain 2 and 3 and ran the NRLB algorithm again using this sequence as input. The result is displayed in Supplementary Fig. 5c and shows that the canonical Dfd-Exd high affinity site is predicted to have more than 100fold higher affinity for the Dfd-Exd complexes than the sites present in the enhancer. So, according to the NRLB algorithm, the sites in the AP2x-377 enhancer are of low-affinity in relation to a known high-affinity site.

Thus, although the NRLB has predicted correctly the existence of Dfd-Exd binding sites present in AP2x-377, it failed to provide a relative affinity accurately that reflected the experimentally determined equilibrium dissociation constants. We don't fully understand the reasons for such discrepancy. The NRLB algorithm has shown to efficiently determine binding sites for several TFs including Ubx-Exd low-affinity sites present in the E3N and 7H *shavenbaby* enhancers (Rastogi et al PNAS, 2018). A possible explanation could lie on how the algorithm has been developed to analyse SELEX data. In these experiments, oligos are used with a 12-nt central core region where all nucleotide permutations are included, and flanked by invariant sequences. These flanking sequences might account for the discrepancy in binding affinities. It may be possible that these regions might contribute to the overall binding of Dfd-Exd. We have included these hypotheses now in the Discussion. In addition, it is not important for this study that these sites were predicted to be low-affinity sites, the surprising finding was that although the Dfd-Exd sites present in the AP2 enhancer are high-affinity, they still provide specificity in target gene regulation in one segment, which is different to previous studies (e.g. Crocker et al, 2015).

#4. I am not convinced that the observed 1.3 fold difference in affinity is the causative variable *in vivo*. Considering that the chromatin state *in vivo* is dramatically different than in the EMSA, alternative possibilities can be readily envisioned. In fact, the lack of *in vivo* binding data (by ChIP etc.) is a weakness of the study.

The 1.3-fold increase in affinity that we previously reported concerned the increase in binding efficiency of Dfd-Exd complexes when we converted the Dfd-Exd site 6 in domain 2 to a class 2 consensus binding site. This was reported in Supplementary Figure 8. This Figure has now been removed and replaced with Supplementary Figure 7 and 8. As described in point #3, we have now characterized in more detail the binding of Dfd-Exd complexes to AP2x-377. We calculated Dfd-Exd equilibrium dissociation constants (Kds) to determine the affinity with which Dfd-Exd complexes bound sites 10 and 11 (located in domain 3 of AP2x-377) and their optimised versions (when these sites were converted to class 2 consensus Dfd-Exd binding sites). These binding sites were chosen because they were shown to be required for the activity of AP2x-377 (Figure 5e-e"). As seen in Supplementary Figures 7 and 8, optimization of these sites resulted in a 2-2.5-fold increase in affinity (Kd {site 10}=16.87 nM, Kd {site 10^{con}}=8.16 nM, Kd{site 11}=16.79 nM and Kd{site 11^{con}}= 6.85 nM). We also confirmed our *in vitro* results, by determining *in vivo* the binding of Dfd to the AP-2 enhancer when binding sites 10 and 11 were converted to class 2 Dfd-Exd high-affinity sites. We have now made a Chip experiment and immunoprecipitated Dfd protein bound to the different enhancer versions *in vivo*. As you can see in Supplementary Fig. 8d, converting sites 10 and 11 in to class 2 Dfd-Exd high-affinity sites resulted in an increase binding of Dfd *in vivo*. These results further support our *in vitro* results and strongly suggest that the ectopic activation observed for AP2x-377-D23H and AP2x-377-D3H are the result of increase binding of Dfd-Exd complexes allowing these enhancers to be active in cells expressing lower levels of Dfd that in wild-type conditions are not sufficient to activate them.

We further tested our model by modifying the levels of Dfd in the maxillary segment. As you may see in the new Figure 7, we use the UAS/Gal4 system to increase the levels of Dfd in the maxillary segment. More specifically we use the AP2-377-D23H-Gal4 driver to increase the levels of Dfd in the anterior region of the

maxillary segment. The increase of Dfd in this region resulted in the ectopic activation of *AP2x-377*. This result supports and complements our initial experiments showing that the increase affinity of Dfd-Exd binding sites in *AP-2* enhancers resulted in ectopic activation in the anterior region of the maxillary segment. This result is very critical, as it also shows that the *AP2x-377* enhancer is not activated in the anterior part of the maxillary segment as the chromatin is closed (as suggested by the reviewer) but can be activated when Dfd levels are increased. We do agree that activation is not that strong, indicating that the chromatin in these cells might be a bit different, thus chromatin could be an issue. We thus wrote on p. 12:

These results showed that the *AP-2* enhancer responds to different Dfd levels in the maxillary segment and is normally unresponsive in anterior maxillary cells due to the low Dfd levels in this region and not (or only a minor extent) due to the chromatin being inaccessible.

Furthermore, using *ptc-Gal4* as a driver, we performed RNAi of Dfd in the anterior part of the maxillary segment to determine if the decrease in levels of Dfd could reduce the activity of *AP2x-377-D23H*. We were able to reduce Dfd levels in the anterior cells of the maxillary segment by 2-fold, resulting in a 2-3-fold reduction of ectopic activity of *AP2x-377-D23H*. Changes in Dfd and GFP reporter expression have been quantified in the respective Figure (Figure 7).

Thus, these results show that either small variations in Dfd-Exd binding affinity or levels in the Dfd result in a loss of domain-specific activation of *AP2x-377* in the maxillary segment. Overall, the new data strongly supports a model where activation of *AP2x-377* in the maxillary segment is the result of a balance between the affinity of the Dfd-Exd binding sites present in the *AP-2* enhancer and the concentration levels of Dfd across the maxillary segment.

#5. The naming of the various domains, binding sites, mutants and oligos is very confusing. It is frequently very difficult to understand how a particular mutation relates to a specific binding site and the corresponding element of the *AP2x* enhancer.

Following the reviewer's suggestions, changes were introduced in the text and in the new figures to simplify that data presented.

REVIEWERS' COMMENTS

Reviewer #1 (Remarks to the Author):

The authors significantly improved their manuscript and met all the comments and questions raised. Important new experiments have been carried out which are now described and discussed in the manuscript (eg, EMSAs with Kd calculations). The authors have also rewritten several parts of the manuscript to better qualify their conclusions, avoiding overstatements and inappropriate generalization of their results.

Overall, this revised manuscript presents significant new data highly relevant to the field, which challenge the current view about how Hox proteins achieve their functional specificity. This deserves publication in a broad-scope journal like Nat Communications.

Reviewer #2 (Remarks to the Author):

All the issues I raised in my initial review have been satisfactorily addressed by the authors. This new version of the manuscript is much improved: it provides stronger evidence and it is more logically organised and easier to follow.

Reviewer #3 (Remarks to the Author):

The authors have dramatically improved their manuscript in several key ways:

1. It is much easier to follow the flow of experiment and the naming convention of domains, oligos, and reporters is much improved.
2. The conclusions of the paper have been toned down so as to not generalize to all anterior hox genes. The authors still draw somewhat provocative conclusions, but I believe they are appropriate based on the data presented and will serve as a useful challenge to the field.
3. The definitions of 'low' and 'high' affinities are much clearer.
4. Key quantitative experiments (determining Kds, doing ChIP) have been included and significantly improve the manuscript.

I believe this manuscript provides novel information of great interest not only to Hox regulation, but to transcription in general. I do have a few remaining comments:

Major concerns (I believe this MUST be addressed before publication):

1. The ChIP experiment in Supp. Fig8d must be clarified further.
 - i) Error bars are shown but it is not clear how many replicates were done (technical versus biological). There is also an asterisk, presumably indicating statistical significance but it is unclear what the p Value is and how it was obtained (since multiple conditions are tested, a corresponding statistical tool should be used).
 - ii) The authors note a 1.4-fold increase in ChIP signal for the D23H construct, but this construct is also expressed in more cells (Fig. 6). Hence the increased ChIP signal may simply be due more cells contributing to the signal (not to more binding at the D23H enhancer in the same number of cells). The authors should correct for the number of cells expressing the reporter and/or discuss this caveat.
2. In general (for instance Fig.6I), the statistical analyses should be clarified further (number of animals, number of cells scored, number of biological replicates, statistical test etc)

Minor concerns (I believe addressing this will improve the manuscript)

1. I still find it disorienting that the initial EMSAs (Fig. 4 and accompanying Supp. Figs) are not compared to the consensus motif. The authors' assertion that the predicted low affinity Dfd-Exd sites are actually 'high' affinity is hard to accept when there is no comparison made to a true high affinity site. I realize that this information is provided in later figures in the form of Kds, but I would urge the authors to include information on the relative affinity of the in vivo sites versus the consensus site already in Fig. 4.

2. I would suggest including some evidence that the NRLB can actually predict a true high affinity site along with Fig. 3. As it stands, the NRLB could just be a bad algorithm. This is covered to some extent in the discussion, but I think including an example of the NRLB algorithm correctly predicting true high affinity sites would give the reader more confidence in its utility up front. Also, does the NRLB program accurately predict low affinity posterior sites?

3. I do not agree that the authors have proven an essential role for the region between elements 2 and 3 based on the data provided. This is not a main point of the paper, but to prove this they should delete it (and/or replace it with an unrelated sequence) in the AP2x-377 construct. Otherwise, I would suggest toning down this conclusion.

Answers to Reviewer 3:

Major concerns (I believe this MUST be addressed before publication):

We have addressed all of the major concerns in the following way:

1. *The ChIP experiment in Supp. Fig8d must be clarified further.*

- i) Error bars are shown but it is not clear how many replicates were done (technical versus biological). There is also an asterisk, presumably indicating statistical significance but it is unclear what the p Value is and how it was obtained (since multiple conditions are tested, a corresponding statistical tool should be used).

We thank the reviewer for noticing. The missing annotations were added to the Figure legend in the supplementary (Fig8d) as well as in the Materials & Methods section. In brief, the ChIP was performed in two replicates and each replicated was tested twice in a qPCR. The four resulting values were used to plot the graph and the resulting arrow bar. The significant difference between the *AP2x-377-wt* and *AP2x-377-D23H* was tested by t-test and results in a p-value = 0.0366, annotated with *. In addition, we performed a Wilcoxon rank test, which results in a p-value of 0.0294, confirming the significance of the difference archived in the two-sided t-test.

- ii) The authors note a 1.4-fold increase in ChIP signal for the D23H construct, but this construct is also expressed in more cells (Fig. 6). Hence the increased ChIP signal may simply be due more cells contributing to the signal (not to more binding at the D23H enhancer in the same number of cells). The authors should correct for the number of cells expressing the reporter and/or discuss this caveat.

We thank the reviewer for his/her comment on that matter. We agree that we cannot exclude the possibility that the increase is (at least in part) due to the increased number of cells expressing the reporter. To clarify this matter, we have included the following sentence in the manuscript (page 10):

However, we can also not exclude the possibility that this (or part of the) increase in Dfd binding is caused by the elevated number of maxillary cells responding to Dfd in the AP2x-377-D23H enhancer context.

2. In general (for instance Fig.6I), the statistical analyses should be clarified further (number

of animals, number of cells scored, number of biological replicates, statistical test etc)

We included all the missing numbers of the counted animals, number of biological replicates in the figure legends and indicated the statistical test used and the corresponding results.

Minor concerns (I believe addressing this will improve the manuscript)

We thank the reviewer for his opinion, here is our view on the points raised.

1. I still find it disorienting that the initial EMSAs (Fig. 4 and accompanying Supp. Figs) are not compared to the consensus motif. The authors' assertion that the predicted low affinity Dfd-Exd sites are actually 'high' affinity is hard to accept when there is no comparison made to a true high affinity site. I realize that this information is provided in later figures in the form of Kds, but I would urge the authors to include information on the relative affinity of the in vivo sites versus the consensus site already in Fig. 4.

We thank the reviewer for his comments. However, we are not sure what the reviewer means by including the information on the relative affinity of the *in vivo* versus the consensus site in Figure 4. In the section “**Dfd-Exd complexes bind low-affinity binding sites in the AP2x-377 enhancer**”, we address the relative affinity of both Dfd-Exd binding sites present in AP2x-377 and of a class 2 Dfd-Exd consensus binding site identified by NRLB algorithm (Figure 3a, Supplementary Figure 5c?). We believe the reviewer suggestion is to address the affinity of Dfd-Exd complexes towards the Dfd-Exd sites present in AP2x-377, as measured by the determination of the Kds, in an earlier part of the paper, specifically in Figure 4. However, at this stage, this suggestion would imply major rewriting of the paper, changing the flow of the story. Furthermore, we do not believe it would significantly improve the flow of information or the overall impact of our findings.

2. I would suggest including some evidence that the NRLB can actually predict a true high affinity site along with Fig. 3. As it stands, the NRLB could just be a bad algorithm. This is covered to some extent in the discussion, but I think including an example of the NRLB algorithm correctly predicting true high affinity sites would give the reader more confidence in its utility up front. Also, does the NRLB program accurately predict low affinity posterior sites?

We thank the reviewer for his suggestion. Concerning the performance of the NRLB algorithm, we do not feel comfortable stating that the NRLB is a bad algorithm. In fact, when we analyse the potential to identify Hox-Exd sites, the NRLB algorithm has performed efficiently. With

exception of two overlapping binding sites present in domain 2, all other predicted sites were bound specifically by Dfd-Exd complexes *in vitro*. As we covered to some extent in the discussion, the differences between the predicted NRLB relative affinity and the affinity as determined by the equilibrium dissociation constant (Kds) could be explained by the use of SELEX experimental results to train the NRLB algorithm. The SELEX experiments searched for Hox-Exd binding sites assuming that it comprises a 12-nucleotide element. Our interpretation, as stated in the paper's discussion, is that additional flanking sequences might play an important role in determining the overall affinity of the Hox-Exd complexes. The ability of the NRLB as an algorithm to identify Hox-Exd binding sites is supported by experimental data. In the study of the *shavenbaby* enhancers *E3N* and *7H*, the NRLB algorithm not only was able to identify previously characterised Ubx low-affinity sites by Crocker *et al.*, 2015 but also extended the study of these enhancers by further identifying and experimentally validating additional low-affinity sites (Rastogi *et al.* 2018). The algorithm was able to identify Scr-Exd binding sites in *fkh250* and *fkh250^{con}* enhancers with a predicted similar relative affinity that was consistent with the previously obtained equilibrium dissociation constants (Joshi *et al.*, 2007, Rastogi *et al.* 2018) as well as to identify class 2 and class 3 high-affinity Hox-Exd sites in the *vvl 1+2* enhancer (Sanchez-Higueras *et al.*, 2019). Furthermore, the algorithm correctly predicted Scr-Exd high-affinity sites in *fj-1* enhancer in a very recent study (Feng *et al.*, 2022). Thus, we assume the algorithm to be able to predict the affinity of many (but maybe not of all) Hox-Exd sites in enhancers.

As for whether the NRLB can faithfully predict the affinity of Hox-Exd binding sites, the answer to that question is not a simple one due to the absence of an in-depth characterisation of the affinity of identified Hox-Exd binding sites and the different experimental conditions between studies where such characterisation was made, specifically the use of full-length proteins or fragments containing only specific domains of those proteins. An example is the binding of Dfd-Exd complexes to the modC Deformed auto-regulatory enhancer. Joshi *et al.*, 2010 characterised the binding of Dfd-Exd to the site I in the modC enhancer and determined its equilibrium dissociation constant (Kd) to be 5.5±0.28 mM. Interestingly, Dfd-Exd has a higher affinity to this site than to a class 2 canonical site. We calculated the Kd of Dfd-Exd to a canonical class 2 Dfd-Exd site to be 9.1 mM (Supplementary Figure 7d). However, these two experiments cannot be compared as the experimental conditions differ. While in our study we used full length Dfd protein in our EMSAs, that was not the case in the study of Joshi *et al.*, 2010 where only the residues 130–586 of Dfd protein were used.

In the end, it does not really matter for our study whether the algorithm can predict the affinity of all Hox-Exd binding sites correctly, as we identified the sites to be of high-affinity by independent EMSA experiments. This is maybe one of the take-home messages, that predictions from the NRLB algorithm should be taken with caution: it can quite accurately

predict Hox-Exd sites but the affinities should be confirmed by independent experiments (like kD measurements in EMSAs). We included this statement in our Discussion (page 15: “*Thus, even though the NRLB algorithm faithfully predicts Hox-Exd sites, the corresponding affinities of these sites should be validated by independent experiments.*”)

3. *I do not agree that the authors have proven an essential role for the region between elements 2 and 3 based on the data provided. This is not a main point of the paper, but to prove this they should delete it (and/or replace it with an unrelated sequence) in the AP2x-377 construct. Otherwise, I would suggest toning down this conclusion.*

As the reviewer rightly states, the role of the non-conserved region between domains 2 and 3 is not the main goal of this paper. Regarding the role of this non-conserved domain, and contrary to the reviewer’s comment, we do not claim this region to be essential for the activity of AP2x-377. Instead, we recognise this element as contributing to the activity of the enhancer in the maxillary segment (which is in line with the reviewer’s request) (page 6: *These results suggested that in addition to the conserved domains, the non-conserved region between these two elements is required for enhancer activity*). The experiments suggested by the reviewer to delete or replace the non-conserved region with an unrelated sequence are interesting and would contribute to a more in-depth understanding of the AP2x-377 enhancer. It is possible that the non-conserved region may provide a “spacer region” between domains 2 and 3 with the resulting architecture having an important role in the overall activity of AP2x-377. However, although we cannot exclude this hypothesis, we do not favour it. We believe our data suggest that the non-conserved region harbours factors that contribute to the activity of AP2x-377. As shown in Figure 2, domains 2 and 3 (AP2x-109 and AP2x-152, respectively) by themselves are unable to drive expression in the maxillary segment (Fig. 2a, h, h’, l and l’). However, when fused to the non-conserved region, the resulting fragments, AP2x-230 (containing domain 2) and AP2x-268 (containing domain 3), were able to drive, albeit incomplete, expression in the maxillary segment (Fig 2a, f, f’, g and g’). Hence based on these results, we recognise that the non-conserved region together with domains 2 and 3 seem to be important for the full maxillary activity of AP2x-377.